# X-ray structure of LeuT in an inward-facing occluded conformation reveals mechanism of substrate release

Kamil Gotfryd [1,2,8], Thomas Boesen [3,4,8], Jonas S. Mortensen[1], George Khelashvili [5], Matthias Quick[6], Daniel S. Terry[7], Julie W. Missel [2], Michael V. LeVine[5], Pontus Gourdon [2], Scott C. Blanchard [7], Jonathan A. Javitch[6], Harel Weinstein [5], Claus J. Loland [1✉], Poul Nissen [3,4✉] & Ulrik Gether [1✉]

Neurotransmitter:sodium symporters (NSS) are conserved from bacteria to man and serve as targets for drugs, including antidepressants and psychostimulants. Here we report the X-ray structure of the prokaryotic NSS member, LeuT, in a $Na^+$/substrate-bound, inward-facing occluded conformation. To obtain this structure, we were guided by findings from single-molecule fluorescence spectroscopy and molecular dynamics simulations indicating that L-Phe binding and mutation of the conserved N-terminal Trp8 to Ala both promote an inward-facing state. Compared to the outward-facing occluded conformation, our structure reveals a major tilting of the cytoplasmic end of transmembrane segment (TM) 5, which, together with release of the N-terminus but without coupled movement of TM1, opens a wide cavity towards the second $Na^+$ binding site. The structure of this key intermediate in the LeuT transport cycle, in the context of other NSS structures, leads to the proposal of an intracellular release mechanism of substrate and ions in NSS proteins.

[1] Molecular Neuropharmacology and Genetics Laboratory, Department of Neuroscience, Faculty of Health and Medical Sciences, University of Copenhagen, Maersk Tower 7-5, DK-2200 Copenhagen N, Denmark. [2] Membrane Protein Structural Biology Group, Department of Biomedical Sciences, Faculty of Health and Medical Sciences, University of Copenhagen, Maersk Tower 7-9, DK-2200 Copenhagen N, Denmark. [3] DANDRITE - Nordic EMBL Partnership for Molecular Medicine, Department of Molecular Biology and Genetics, Aarhus University, DK-8000 Aarhus C, Denmark. [4] Interdisciplinary Nanoscience Center iNANO, Aarhus University, DK-8000 Aarhus C, Denmark. [5] Department of Physiology and Biophysics, Weill Cornell Medical College, Cornell University, New York, NY NY10065, USA. [6] Department of Psychiatry, Columbia University Vagelos College of Physicians & Surgeon and Division of Molecular Therapeutics, New York State Psychiatric Institute, New York, NY 10032, USA. [7] Department of Structural Biology, St. Jude Children's Research Hospital, Memphis, TN 38105, USA. [8] These authors contributed equally: Kamil Gotfryd, Thomas Boesen. ✉email: cllo@sund.ku.dk; pn@mbg.au.dk; gether@sund.ku.dk

Neurotransmitter:sodium symporters (NSS) constitute a large family of Na⁺-coupled, secondary active transmembrane transporters conserved across all kingdoms of life that includes carriers for a broad variety of solutes, including amino acids and neurotransmitters[1,2]. Key mammalian NSS members include the transporters for dopamine (DAT), γ-aminobutyric acid (GAT-1-4), glycine (GlyT1-2), norepinephrine (NET), and serotonin (SERT) that serve central roles in clearing released neurotransmitters from the extracellular space in the brain[3,4]. Malfunctions of NSS proteins play important roles in many pathophysiological conditions, including attention deficit hyperactivity disorder (ADHD), anxiety, depression, epilepsy, and schizophrenia[3–5]. Concurrently, NSS are targets for several medicines (e.g., ADHD medication, antidepressants, antiepileptics) and for drugs of abuse, including amphetamines, cocaine, and MDMA (ecstasy)[4].

NSS proteins transport solutes by an alternating access mechanism[6], based on transitions between outward- and inward-facing conformations that leave bound substrates accessible to the extracellular or intracellular environment, respectively[7–9]. The transition from outward- to inward-facing conformations associated with solute transport is driven by the transmembrane Na⁺-gradient[7–9]. The first insight into the atomic structure of NSS proteins was achieved by the X-ray structure of the leucine transporter LeuT, a prokaryotic NSS member from the thermophilic bacterium *Aquifex aeolicus* (PDB-ID: 2A65)[10]. The transporter was captured in a Na⁺/substrate-bound, outward-facing occluded state and the structure confirmed a predicted topology of 12 transmembrane segments (TMs). In addition, it revealed a structural fold with the first ten TMs arranged in a pseudosymmetric inverted-repeat pattern with a central primary substrate binding site and two sodium binding sites (named Na1 and Na2) in close proximity[10]. Additional insights into the structural underpinnings of transport by NSS proteins were obtained by crystallization of other transport cycle intermediates of LeuT, including a Na⁺-bound, outward-facing open state (PDB-ID: 3TT1)[11], an apo inward-facing open state (PDB-ID: 3TT3)[11] and an apo outward-facing occluded return state (PDB-ID: 5JAE)[12]. Moreover, another bacterial NSS member, MhsT, has been crystallized in two Na⁺/substrate-bound, inward-facing occluded states (PDB-IDs: 4US3 and 4US4)[13], and structures have been obtained for the *Drosophila melanogaster* dopamine transporter (dDAT)[14,15] and for the human SERT (hSERT)[16,17]. Importantly, the structural fold appears highly conserved throughout the NSS family, as the structures of dDAT[14,15] and hSERT[16,17], as well as of MhsT[13], display remarkably high similarity to the structures of LeuT[10].

One of the outstanding mechanistic questions is how binding of the extracellular substrate couples to the opening of the inner gate and the subsequent release of substrate and ions to the intracellular environment. Although LeuT has been crystallized in several different conformations, we still lack critical molecular level information about conformational transitions in LeuT along the path towards intracellular substrate release. The process has been studied with computational approaches[12,18–23] and a broad variety of biophysical and biochemical techniques[24–34]. Indeed, results from single-molecule fluorescence resonance energy transfer (smFRET), spin labeling and site-directed fluorescence quenching spectroscopy (SDFQS) studies support the existence of distinct conformations on the trajectory from the outward-facing occluded state to the full apo inward-facing open conformation[25,28,29,35]. Furthermore, the existence of such intermediate conformations has been confirmed by structures of inward-facing occluded states of MhsT[13].

Here we present the X-ray structure of LeuT in a Na⁺/substrate-bound, inward-facing occluded conformation at 2.6 Å resolution. In the search for such a state, we exploited results obtained from smFRET indicating that L-Phe binding induces an inward-facing open state of LeuT to a much higher extent than L-Ala. In parallel, molecular dynamics (MD) simulations and functional characterization suggested that Ala substitution of the conserved N-terminal Trp8 (W8A) triggers opening toward the intracellular environment. The W8A mutation combined with L-Phe binding enabled crystallization and structure determination of a state differing from previously determined conformations. As compared to the outward-facing occluded state of LeuT (PDB-ID: 2A65)[10], the structure shows a pronounced outward tilting of the cytoplasmic end of TM5 that, together with a rearrangement of the N-terminus, but without coupled movement of TM1, opens a wide cavity toward the Na2 site. Moreover, in this construct the sodium ions in both sites have moved toward the cytoplasm with the Na2 sodium ion showing a distorted coordination geometry. This structure complements currently available information on the NSS transport mechanism by revealing an occluded inward-facing conformation that precedes the solvation of the Na2 site, which in turn serves as a trigger for the full release of the Na1 sodium ion and the substrate.

## Results

**L-Phe shifts LeuT_WT toward an inward-facing intermediate.** Application of smFRET to LeuT has enabled quantitative measurements of intracellular occlusion-opening dynamics by providing a means to detect time-dependent intramolecular distance fluctuations at the single-molecule level[26–28]. As previously revealed[26–28], LeuT labeled with donor and acceptor fluorophores at the cytoplasmic interface exhibits both high- and low-FRET efficiency in the absence of Na⁺ and substrate, which have been assigned as corresponding to inwardly occluded and open conformations, respectively. Moreover, it was found that saturating Na⁺ concentrations in the absence of substrate stabilized the high-FRET, inwardly occluded state (FRET efficiency ~0.8) while the addition of L-Ala in the presence of a low concentration of Na⁺ (5 mM) markedly increased the occupancy of an intermediate-FRET state (FRET efficiency ~0.6), which has been associated with a partially open, inward-facing conformation[28]. It was also shown that this intermediate conformation could be converted to the high-FRET, inwardly occluded state in the presence of high Na⁺ concentrations relative to the EC₅₀. On that basis, it was hypothesized that the intermediate-FRET state corresponds to an inwardly open intermediate of LeuT prior to full inward opening[28].

We decided to test whether such a conformation of LeuT could be selectively induced by substrates other than L-Ala. To our surprise, we found that L-Phe, in the presence of 5 mM Na⁺, increased the occupancy of an intermediate-FRET state (FRET efficiency ~0.6; Fig. 1a, b) to a much greater extent than previously shown for L-Ala[28] with an EC₅₀ of ~5 μM (Fig. 1c). Interestingly, the dominant effects in the presence of L-Phe at concentrations up to 1 μM (and thus at low concentrations relative to the EC₅₀) were a ~4-fold increase in the rate of transitioning from the low- to intermediate-FRET state and a ~2-fold increase in the rate of transitioning from the intermediate- to the low-FRET state (Fig. 1d). By contrast, at saturating concentrations of L-Phe, the dwell times in the low- and intermediate-FRET states became so short that they could not be resolved within the current experimental time resolution (25 ms), which resulted in what appeared to be relatively stable dwells in the intermediate-FRET state with minimal dynamics (Fig. 1a, bottom panel). This behavior is in marked contrast to that of L-Ala, where transitions between intermediate- and high-FRET states are present even at saturating concentrations[27]. From these data, we concluded that

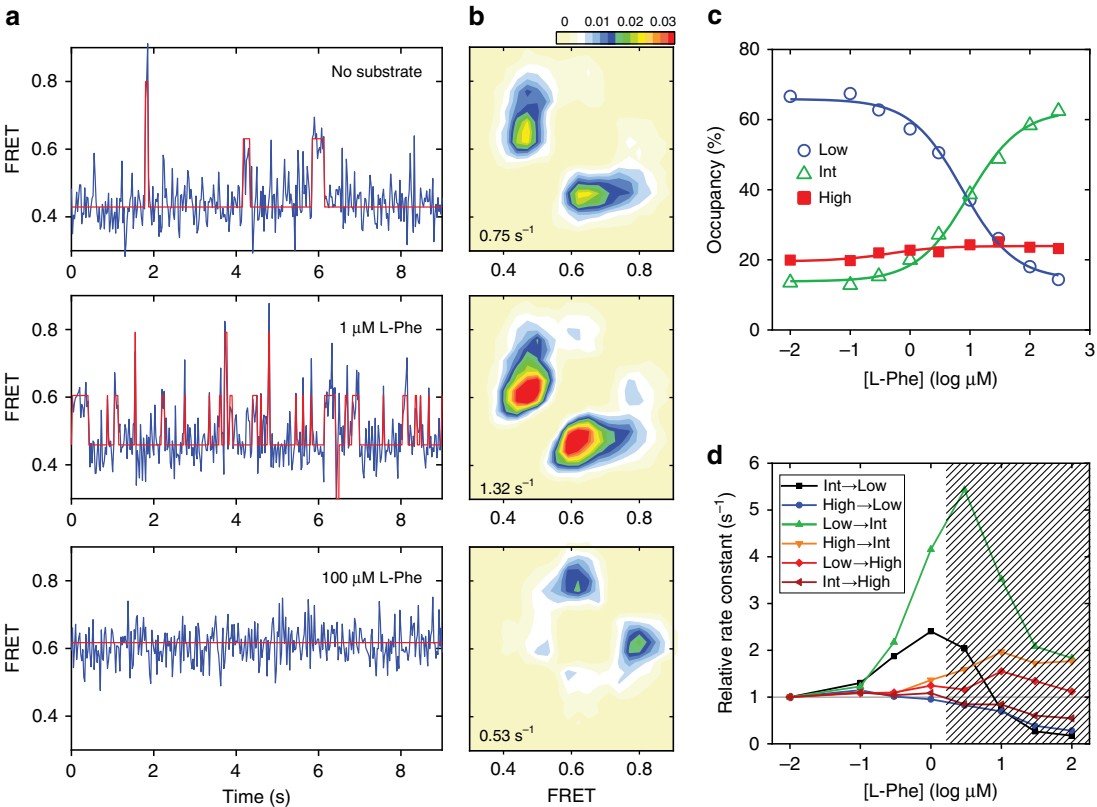

**Fig. 1 Effect of ʟ-Phe on LeuT$_{WT}$ dynamics. a** Intracellular-labeled LeuT$_{WT-7C/86C}$ was imaged with single-molecule fluorescence resonance energy transfer (smFRET) in the presence of 5 mM Na$^+$ and in the absence of substrates (top), in the presence of 1 μM ʟ-Phe (middle) or in the presence of 300 μM ʟ-Phe (bottom). Shown are representative smFRET traces (blue) with state assignment traces (red). **b** Transition density contour plots showing the mean FRET value in the dwell just prior to (bottom axis) and just after (left axis) each transition between distinct FRET states (high-FRET state with a FRET efficiency of ~0.8, intermediate-FRET state with a FRET efficiency of ~0.6 and low-FRET state with a FRET efficiency of ~0.4), summed into a histogram. The average transition rate between states is given at the bottom left of each panel. Transitions per second rate scale is shown on top. **c** Ensemble-averaged occupancy in the low- (blue circles), intermediate- (green triangles) and high-FRET states (red squares) were fitted to dose-response functions with the Hill coefficients fixed at 1.0 and EC$_{50}$ of 5.1 μM (lines). **d** Rate constants for each transition type from experiments conducted in the presence of the indicated concentration of ʟ-Phe (bottom axis), relative to the absence of substrate. Shaded area indicates regime in which time averaging obscures transitions, resulting in erroneous rate estimates. Source data are provided as a Source Data file.

ʟ-Phe binding to LeuT results in an apparent stabilization of a functionally relevant, inward-open conformation on the pathway to intracellular release of Na$^+$ prior to full inward opening.

**Mutation of Trp8 shifts LeuT toward an inward-facing state.** We hypothesized that the introduction of a mutation may complement ʟ-Phe stabilization to further enrich inward-open states. As the N-terminus assists in establishing Na$^+$/substrate occlusion and is not visible in the apo inward-facing open state (PDB-ID: 3TT3)[11], we decided to focus on disrupting its interaction with the transmembrane domain. In previous computational simulations of the outward-open conformation, the highly conserved tryptophan residue at position 8 of LeuT (W8; Fig. 2a and Supplementary Fig. 1) was found to restrain the N-terminus in a stabilizing network of interactions with Y265 and Y268 (Fig. 2a) that blocks the access to the substrate binding site from the cytoplasm[36]. We therefore introduced the W8A mutation in the outward-facing occluded structure of LeuT (PDB-ID: 3GJD; W8A$_{OUT-OCC}$) and investigated its conformational dynamics with unbiased ensemble MD simulations. As a control, we also simulated the 3GJD-based wild type (WT) LeuT model (WT$_{OUT-OCC}$) in such ensemble MD protocols. The conformations of W8A$_{OUT-OCC}$ collected from the ensemble trajectories were projected onto two-dimensional space of the distances of the collective variables[20,37] that represent the functional intracellular gates in LeuT, including the distances

between R5 and D369, and between Y268 and Q361 (Fig. 2b and c for W8A$_{OUT-OCC}$ and WT$_{OUT-OCC}$, respectively). In Fig. 2b, c, each black dot represents one snapshot from the combined trajectories of the replicas while the model snapshots of the inner gate show structural representations of the selected high population states (termed microstates and annotated "a, b, c"). The analysis revealed that the WT$_{OUT-OCC}$ system (Fig. 2c) visited states with various degrees of separation between R5 and D369 (Fig. 2c, microstates "a–c" span a large range of R5-D369 distances). These include the closed R5-D369 gate (~4 Å distance, microstate "a") seen in the outward-facing occluded structure of LeuT (PDB-ID: 3GJD), as well as R5-D369 distances that are larger (microstates "b–c"). These diverse conformations of the R5-D369 gate correspond to the various modes in which the N-terminus positions with respect to the transmembrane bundle (see structural snapshots in Fig. 2c). However, in all these states the Y268-Q361 gate remained closed (as seen from the relatively small range of Y268-Q361 distances; see also Supplementary Fig. 2). In contrast, the W8A$_{OUT-OCC}$ system sampled populations of states in which both the R5-D369 and Y268-Q361 gates were closed (as represented by the initial outward-occluded state model, Fig. 2b, microstate "a"), as well as states in which either one of the gates was opening (Fig. 2b, microstates "b" and "c"; and Supplementary Fig. 2a, b). The observed increase of the Y268-Q361 distance relates to repositioning of the Y268 side chain from facing "into" the protein to

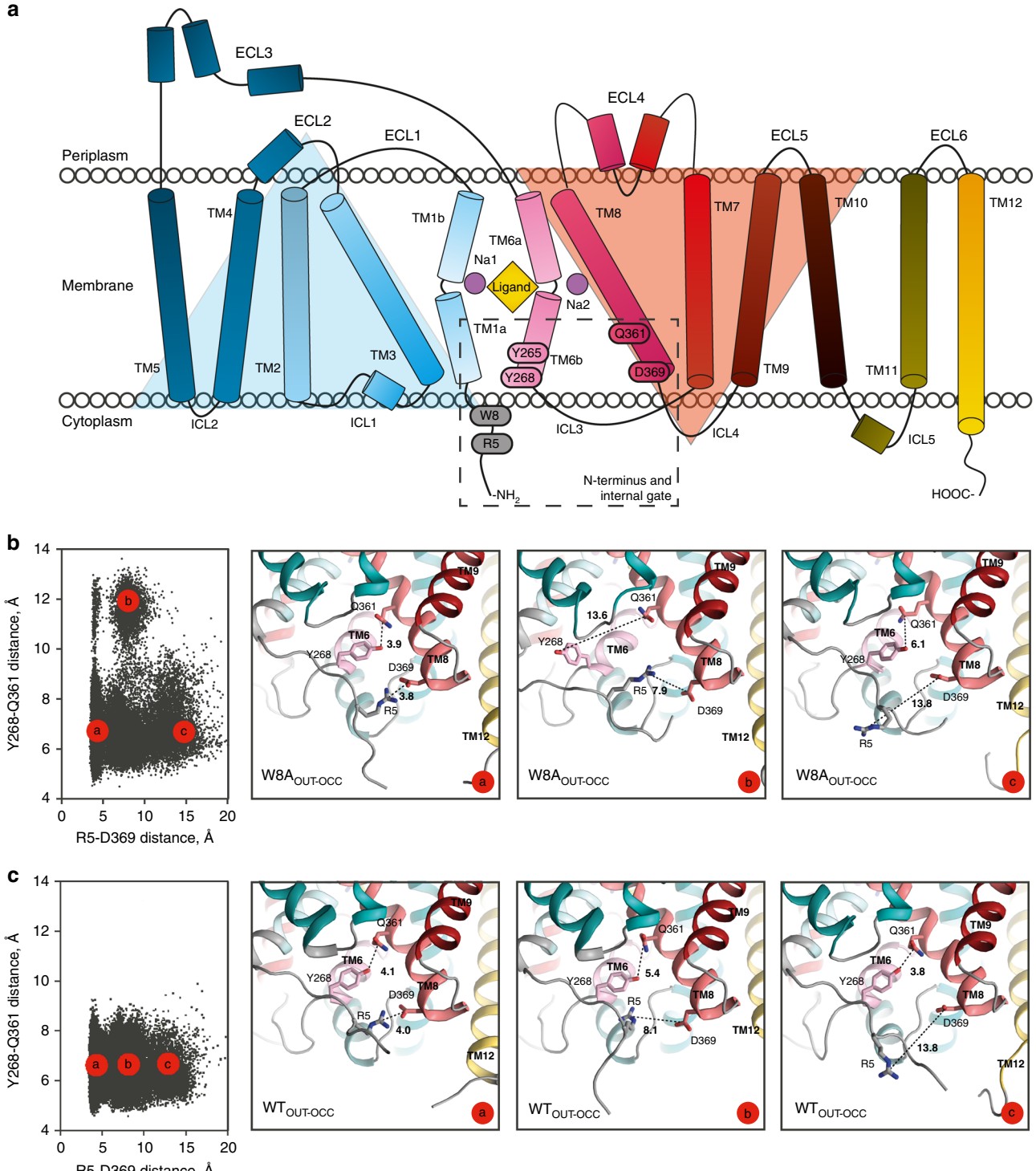

**Fig. 2 Secondary structure of LeuT and conformational dynamics of the intracellular gate. a** Topology of the LeuT monomer with transmembrane segments (TMs) 1-12, extracellular loops (ECLs) 1–6 and intracellular loops (ICLs) 1–5. TMs 1–5 and TMs 6–10 are related by a pseudo two-fold rotation as highlighted by the light blue and orange triangles, respectively[10]. Dashed rectangular indicates region, including the N-terminal W8 residue and residues constituting the internal gate (R5, Y265, Y268, Q361, and D369). **b** Conformations of W8A$_{OUT-OCC}$ LeuT construct collected from the ~10-μs ensemble MD trajectories (run in 10 statistically independent replicates, each ~1-μs long) projected onto two-dimensional (2D) space of the R5-D369 and Y268-Q361 distances. Each black dot (left panel) represents one snapshot from the combined trajectories of the 10 replicas, collected at 800-ps intervals for a total of >10.000 points. The three right panels are structural representations of the system corresponding to the three microstates ("a"–"c"), highlighting various degrees of opening of the two gates. The structures show the protein (close-up of the gate residues R5, D369, Y268, and Q361) with the above-mentioned distances indicated. **c** Conformational dynamics of the intracellular gate of WT$_{OUT-OCC}$ LeuT construct. Both the 2D and corresponding structural representations are shown similarly to Fig. 2b.

facing "out" (compare "b" with "a" and "c" in Fig. 2b). Indeed, the quantification of the $\chi_1$ dihedral angle of Y268 in W8A$_{OUT-OCC}$, but not in WT$_{OUT-OCC}$ (Supplementary Fig. 2c, f), showed that in the trajectories in which Y268 breaks away from Q361, the side chain of Y268 samples "out" conformations ($\chi_1$ values ~ 180 °).

Water penetration from the intracellular side is an early determinant of substrate translocation, as suggested by the crystal structures of MhsT, and several studies of LeuT and other NSS proteins[13,19–21]. Importantly, the first functional step in the transport process, i.e., the release of Na$^+$ from Na2 to the intracellular environment, has been shown in a quantitative kinetic model for hDAT to be allosterically coupled to water penetration through reconfigured intracellular gates[20,21,38]. We therefore quantified the water molecule count in the channel connecting the intracellular vestibule to the Na2 site in the MD simulation trajectories of W8A$_{OUT-OCC}$ and WT$_{OUT-OCC}$, and found stronger water penetration to the Na2 site of W8A$_{OUT-OCC}$ (Supplementary Fig. 3a, b) with half of the trajectories in the W8A$_{OUT-OCC}$ sample states where the channel is populated by on average 6 water molecules (Supplementary Fig. 3c). In comparison, the peak matching the same water count in the WT$_{OUT-OCC}$ construct is substantially lower (Supplementary Fig. 3, red line and Fig. 2c), as it corresponds to only one trajectory displaying a transiently formed water wire (Supplementary Fig. 3d). These results agree

with our previous finding that water wires in this region are transient on the microsecond scale, even when mutations that stabilize the inward-facing state are introduced[35]. Hence, the computational analyses suggest that W8A favors changes in the intracellular gates that give rise to conformations that allow water penetration to the Na2 site.

**LeuT$_{W8A}$ displays distinct transport and binding activities.** Since the W8A mutation appears to shift LeuT toward inward-facing conformations, we anticipated that the substitution would have functional consequences. First, we tested the transport capacity of LeuT$_{W8A}$ upon reconstitution into proteoliposomes using L-[3H]Ala, which is known to be the best substrate for LeuT[39]. While the reconstituted LeuT$_{WT}$ displayed clear time-dependent L-[3H]Ala uptake, no transport was observed for LeuT$_{W8A}$ (Fig. 3a). When testing L-[3H]Phe uptake, however, no transport activity was measured for LeuT$_{WT}$, while very small, yet detectable, uptake was observed for LeuT$_{W8A}$ (Fig. 3b). Using a scintillation proximity binding assay (SPA), we assessed binding affinities for the two substrates, L-Leu and L-Phe, using detergent-solubilized protein samples. For LeuT$_{W8A}$, we observed specific binding of L-[3H]Leu, but the affinity was dramatically decreased (~80-fold; 1630 ± 210, $n = 5$ vs. 20.4 ± 6.3 nM, $n = 3$; means ± S.E., respectively; Fig. 3c). These findings are consistent with the

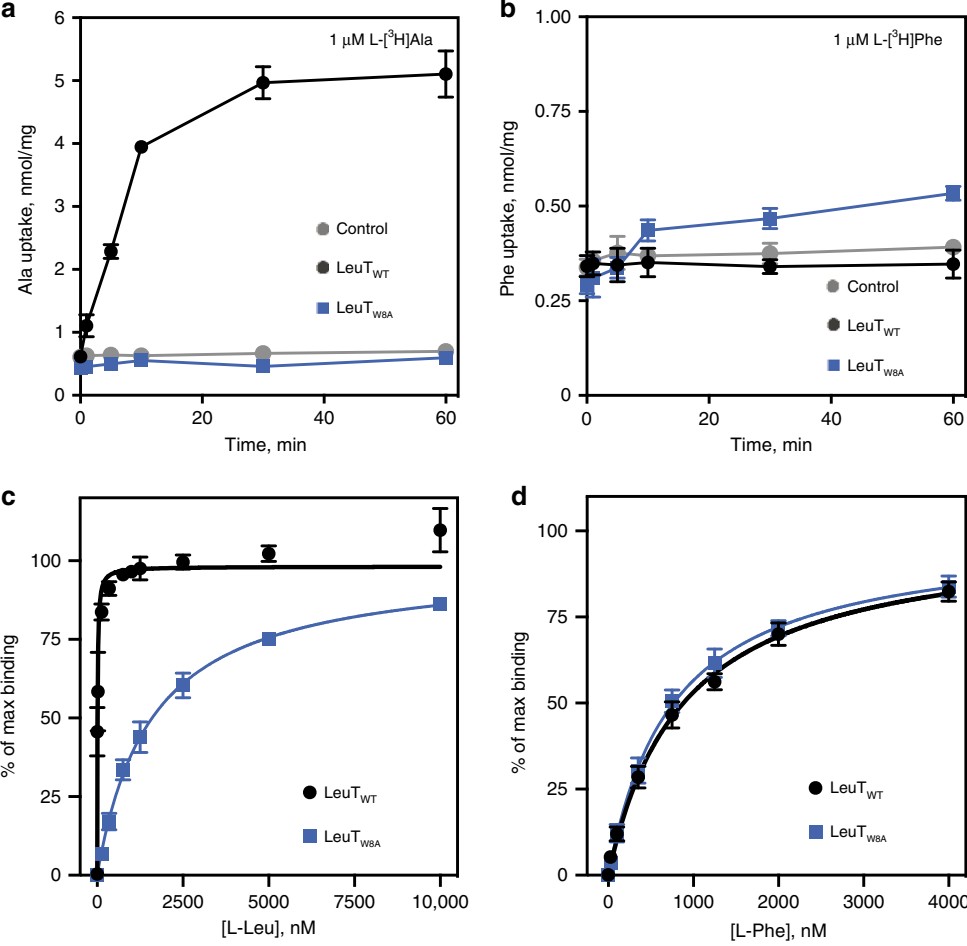

**Fig. 3 Comparison of functional properties of LeuT$_{W8A}$ and LeuT$_{WT}$. a** Time course of L-[3H]Ala uptake in proteoliposomes containing LeuT$_{WT}$ (black circles) or LeuT$_{W8A}$ (blue squares) or in control liposomes (gray circles). **b** Time course of L-[3H]Phe uptake in proteoliposomes containing LeuT$_{WT}$ (black circles) or LeuT$_{W8A}$ (blue squares) or in control liposomes (gray circles). Data in **a**, **b** are means ± S.E, $n = 2, 3$. **c**, **d** Saturation binding of L-[3H]Leu (**c**) or L-[3H]Phe (**d**) performed in 200 mM NaCl using a scintillation proximity assay (SPA) on purified LeuT$_{WT}$ (black circles) or LeuT$_{W8A}$ (blue squares) in detergent solution. Data were fitted to a one site-specific binding model and are shown as percentage of max binding for each construct. All SPA results are means ± S.E. from $n = 3–5$ independent experiments performed in triplicates. Source data are provided as a Source Data file.

shift of the W8A mutant towards a more inward-facing conformation that would result in lower affinity for the substrate. In contrast to our findings for L-Leu, L-[³H]Phe saturation binding experiments (Fig. 3d) revealed a similar affinity for LeuT$_{W8A}$ as compared to LeuT$_{WT}$ ($K_D = 796 \pm 139$ nM, $n = 3$ vs. $K_D = 898 \pm 1128$ nM, $n = 4$, means $\pm$ S.E., respectively). The unchanged affinity of L-Phe in comparison to the dramatic loss of L-Leu affinity for LeuT$_{W8A}$ is consistent with L-Phe promoting an intermediate in the transport cycle that is also favored by the W8A mutation.

**The structure of Na⁺/L-Phe-bound LeuT$_{W8A}$.** We hypothesized that stabilization of LeuT$_{W8A}$ with L-Phe, which binds with slightly higher affinity to W8A than to WT, would help to capture the transporter in an elusive conformational intermediate between the Na⁺/substrate-bound, outward-oriented occluded state and the apo inward-facing open conformation. Indeed, crystals of W8A were obtained for protein samples prepared in the presence of Na⁺ and L-Phe, diffracting X-rays to 2.6 Å resolution (Fig. 4, Supplementary Fig. 4 and Table 1). The structure was determined using molecular replacement (MR) and through screening of all available LeuT structures as search models. The best MR solution was obtained with the apo inward-facing open structure of LeuT (PDB-ID: 3TT3)[11] as the search model yielding R$_{work}$.R$_{free}$⁻¹-values of 21.1 and 23.4, respectively for the final, refined model (Table 1). Clear electron densities were observed for residues 13–511 in the chain A and residues 13–507 in the B chain of the LeuT dimer, but no visible electron density was found for the first 12 N-terminal residues, suggesting that the N-terminus is disordered in the crystals (Supplementary Fig. 4). Interestingly, inspection of the final LeuT$_{W8A}$ model revealed an inward-facing occluded conformation of LeuT (Figs. 4 and 5).

Both Na⁺ ions and the L-Phe substrate are located in their binding sites with an overall coordination chemistry similar to that of the outward-occluded state (Figs. 4 and 5c). The sodium ion in Na1 is octahedrally coordinated, interacting directly with the substrate, as in the outward-occluded state. In Na2, the coordination of the bound Na⁺ ion remains trigonal bipyramidal, also as seen in the outward-occluded state (Fig. 4b and Supplementary Fig. 5), but the coordination is distorted by shifts of TM1 (located between Na1 and Na2) and TM8 (Fig. 5c and Supplementary Fig. 5). Overlaying the L-Phe substrate with L-Leu in the outward-facing occluded structure shows that they superimpose almost seamlessly in the binding pocket (Fig. 5c and Supplementary Fig. 5). The space for the additional atoms in the aromatic ring of L-Phe is provided by rearrangement of the hydrophobic side chains lining the binding pocket that are also involved in the L-Leu binding, demonstrating how LeuT can accommodate different substrates in agreement with previous findings[39] (Fig. 5 and Supplementary Fig. 5). In the binding pocket, F259 forms a π-π stacking with the aromatic ring of the bound L-Phe substrate and F253 in TM6. V104 and I359 are also part of the hydrophobic pocket containing the hydrophobic side chain of the L-Phe substrate, and the conserved Y108 on TM3 forms a hydrogen bond to the carboxylic acid of the bound substrate (Fig. 4b).

Interestingly, the lack of visible electron density for the first 12 N-terminal residues, and the inferred disordering of this segment in the crystals, could be indicative of lost interactions with the cytoplasmic parts of TM6 and TM1a. Consistent with this notion, Y268, which forms a cation-π interaction with R5 in the outward-facing occluded state, has moved into hydrogen bonding distance of D369 (Fig. 4a, c and 5a, b). Furthermore, the critical salt bridge between R5 and D369 is broken. This salt bridge constitutes an important component of the internal gate in the outward-facing occluded conformation that prevents opening of the cytoplasmic

access to the substrate and sodium ion binding sites (Fig. 4a, c and 5a, b).

Relative to the outward-facing occluded state, the putative release of the N-terminal tail and disruption of the R5-D369 interaction are accompanied by a conformational rearrangement of TM5, which corresponds with that observed for the apo inward-facing open structure[11]. This TM5 movement tilts the cytoplasmic end of TM5 away from TM1 (Figs. 4c and 5d-f). Furthermore, in going from the outward-facing occluded to the inward-facing occluded state, intracellular loop (ICL) 2, connecting TM4 and 5, transitions from a compact to a highly extended structure (Fig. 5 and Supplementary Movies 1 and 2). The helical structure of TM5 itself is generally similar between the outward-facing occluded state, the apo inward-facing open state, and the structure here of an inward-facing occluded conformation; but the apo inward-facing open structure shows a partial unwinding of the first N-terminal residues of TM5 (Fig. 5).

The transition to the inward-facing occluded state appears to provide solvent access to the Na2 site, and release of the N-terminus promotes dynamics via TM5 translocation that may lead to the subsequent dissociation of Na⁺ from the Na2 site. Similarly to the apo inward-facing open state[11], the extracellular vestibule is also closed in the LeuT$_{W8A}$ structure, and the extracellular path to the ion and substrate binding sites is closed by concerted movements of the extracellular loops (ECLs) 2, 3, and 4 relative to the scaffold domain (TM3-4 and TM8-9), of which the latter represents a stable structural element that displays no or only limited conformational changes during the transport cycle (Figs. 4 and 5 and Supplementary Movie 3). Importantly, the pull on TM5 by ECL2 and ECL3, moving as a rigid body combined with extension of the ICL2, opens a water permeation pathway towards the Na2 site from the cytoplasmic side (Supplementary Movies 1, 2, and 4). This is further supported by findings from ensemble MD simulations performed on the two chains of the LeuT$_{W8A}$ crystal structure (W8A$_{chainA}$ and W8A$_{chainB}$, respectively). That is, the analysis of the hydration in the intracellular channel of W8A$_{chainA}$ and W8A$_{chainB}$ constructs from the MD trajectories revealed a peak corresponding to the same high-water count level (~6 waters in the channel) (Supplementary Fig. 6), as observed in our MD simulations of the W8A$_{OUT-OCC}$ construct (Supplementary Fig. 3). Importantly, for the W8A$_{chainA}$ and W8A$_{chainB}$ systems this peak has a higher probability of occurrence, which suggests that the W8A mutation does indeed promote an increase in the channel hydration that is likely to ultimately lead to Na2 release. To further quantitatively compare the extent of opening of the intracellular vestibule in the W8A structure to that of the outward-facing occluded state, we calculated the number of water molecules in the intracellular vestibule (defined identically for the two structures) during the course of MD simulations of the respective systems. The volume of the intracellular vestibule in the W8A structure was found to fit on average ~30 waters, whereas in the outward-facing occluded state, on average, only ~10 water molecules populate this region (Supplementary Fig. 7).

We next compared the LeuT$_{W8A}$ structure with that of the bacterial NSS homolog, MhsT, which has been captured in two different Na⁺/substrate-bound inward-facing occluded states (MhsT$^{HILIDE}$, PDB-ID: 4US3, and MhsT$^{LCP}$, PDB-ID: 4US4)[13]. Similar to the present LeuT$_{W8A}$ structure and the apo inward-facing open state of LeuT (PDB-ID: 3TT3)[11], the extracellular portions of both MhsT structures adopt a configuration with a closed extracellular vestibule (Fig. 6)[13]. The cytoplasmic side, however, reveals distinct features in the two different structures as compared to the LeuT$_{W8A}$ structure. In MhsT$^{HILIDE}$, the N-terminus is visible and caps the substrate release pathway by binding to the cytoplasmic surface of the transmembrane domain through interactions that include the conserved tryptophan

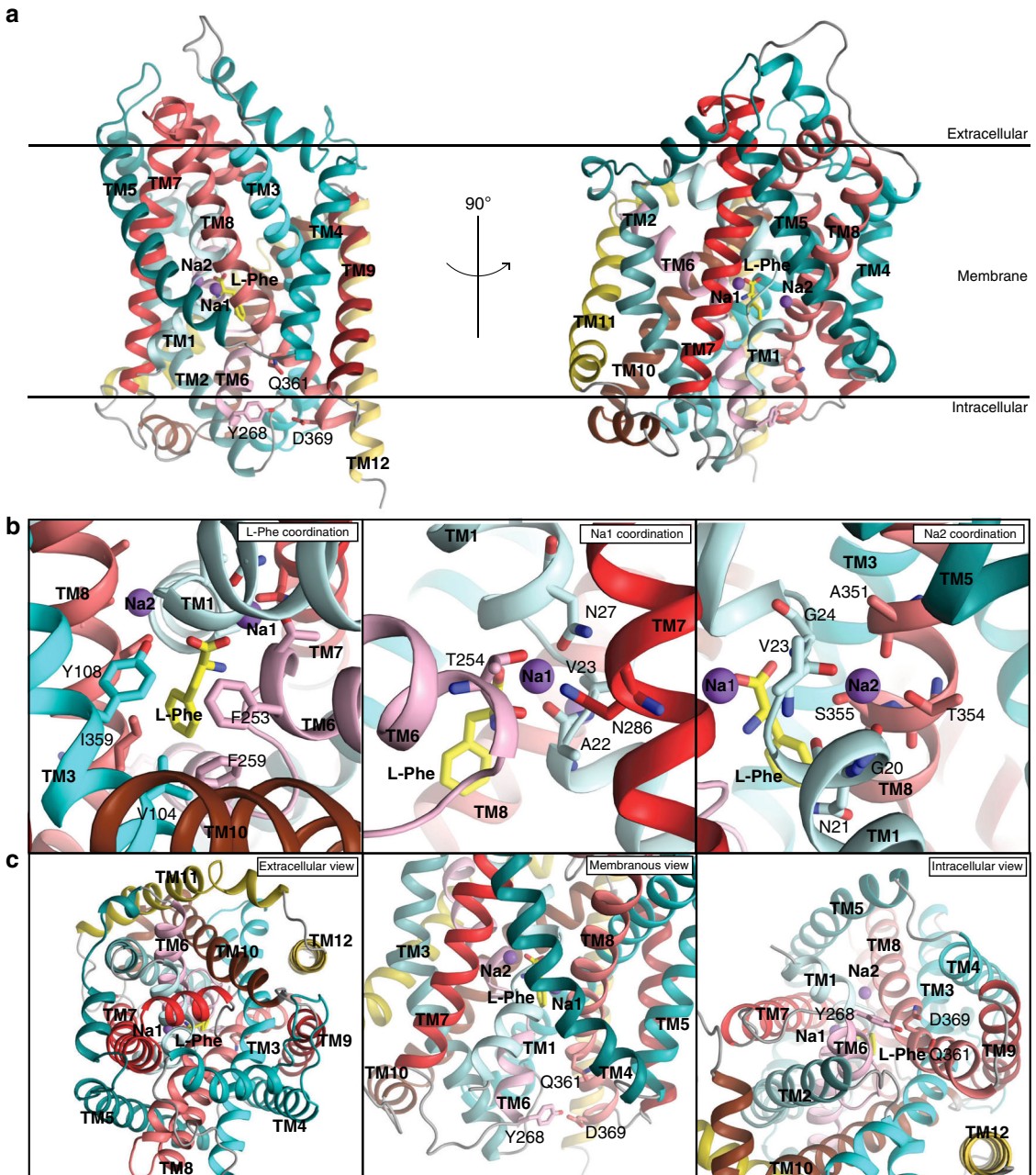

**Fig. 4 Crystal structure of Na$^+$/L-Phe-bound LeuT$_{W8A}$. a** Side views of the LeuT$_{W8A}$ monomer seen along the plane of the lipid bilayer with two sodium ions (purple spheres) and a single L-Phe molecule (yellow sticks) bound centrally. Membrane orientation is indicated. The solved LeuT$_{W8A}$ structure revealed an inward-facing occluded conformation. **b** Close-up view of the Na$^+$/L-Phe binding sites visualizing the residues coordinating binding of L-Phe (left), Na1 (middle) and Na2 (right). L-Phe binds similarly to L-Leu in the outward-facing occluded structure[10]. The space for the additional atoms in the aromatic ring of L-Phe is provided by the hydrophobic side chains lining the binding pocket. F259 forms a π-π stacking with the aromatic ring of the bound L-Phe substrate and F253 on TM6. V104 and I359 are also part of the hydrophobic pocket containing the hydrophobic side chain of the L-Phe substrate while the conserved Y108 on TM3 forms a hydrogen bond to the carboxylic acid of the bound substrate. Na1 displays octahedral coordination by A22, N27, T254, and N286 residues, and interacts directly with L-Phe. Na2 is trigonal bi-pyramidal, and established by interaction with G20, V23, A351, T354, and S355 residues (labeled and shown as colored sticks). **c** Close-up views of the extracellular vestibule region presented from the extracellular side (left), the membranous side (middle) and the intracellular vestibule region seen from the intracellular side (right). Internal gate residues (Y268, Q361, and D369) are labelled and shown as colored sticks. Helices are colored following the same scheme as in the schematic illustration of LeuT in Fig. 2.

equivalent to W8 in LeuT (Fig. 6a–c)[13], similarly to what is observed in the outward-facing occluded state of LeuT (PDB-ID: 2A65)[10]. Moreover, in MhsT$^{HILIDE}$ the N-terminus appears to stabilize an unwound cytoplasmic end of TM5 from P181 (P200 in LeuT) to G190 (Fig. 6)[13]. In MhsT$^{LCP}$, however, the N-terminal segment is disordered, as in LeuT$_{W8A}$ and in the apo

inward-open state of LeuT[11], and TM5 is helical (Fig. 6d–f)[13]. In the LeuT$_{W8A}$ structure, TM5 is helical as well (like in all other structures of LeuT), but, together with ICL2, markedly more peripherally positioned as compared to both MhsT$^{HILIDE}$ and MhsT$^{LCP}$ (Fig. 6)[13]. As a result, the LeuT$_{W8A}$ structure with bound Na$^+$/L-Phe displays a configuration with an enlarged

**Table 1 Data collection and refinement statistics (molecular replacement).**

| | LeuT$_{W8A}$ |
|---|---|
| Data collection | |
| Space group | P1 |
| Cell dimensions | |
| a, b, c (Å) | 77.85, 83.99, 86.94 |
| α, β, γ (°) | 65.38, 89.77, 64.56 |
| Resolution (Å) | 2.6 (2.69–2.60)[a] |
| R$_{sym}$ or R$_{merge}$ | 20.1 (97.3) |
| I / σI | 15.6 (1.22) |
| Completeness (%) | 98.7 (98.1) |
| Redundancy | 6.2 (6.1) |
| Refinement | |
| Resolution (Å) | 43.1–2.6 |
| No. reflections | 53651 (5345) |
| R$_{work}$/R$_{free}$ | 21.1/23.4 |
| No. atoms | 7925 |
| Protein | 7899 |
| Ligand/ion | 4 |
| Water | 22 |
| B-factors (Å$^2$) | |
| Protein | 81.2 |
| Ligand/ion | 72.4 |
| Water | 58.3 |
| R.m.s. deviations | |
| Bond lengths (Å) | 0.002 |
| Bond angles (°) | 0.49 |

Values in parentheses are for highest-resolution shell.
[a]Data were collected from a single crystal.

cytoplasmic cavity that is clearly wider than that seen in either of the MhsT structures, although not as wide as in the fully open conformation seen in the apo inward-facing open state of LeuT (PDB-ID: 3TT3) (Fig. 7 and Supplementary Fig. 8)[11].

## Discussion

LeuT represents a principal structural model for studying the molecular function of NSS proteins[40–42]. Importantly, the structures of the eukaryotic dDAT[14,15] and hSERT[16,17] have confirmed an overall high degree of architectural conservation through evolution, substantiating the validity of exploiting bacterial homologs, such as LeuT, to elucidate fundamental principles underlying transmembrane ion-coupled transport mediated by NSS members[41]. In this study, we determined the structure of a conformational intermediate in the LeuT transport cycle corresponding to an inward-facing occluded, Na$^+$/substrate-bound state (Fig. 7). Our crystallographic approach was guided by smFRET experiments, mutagenesis and computational simulations. Specifically, we took advantage of mutating an almost universally conserved N-terminal tryptophan residue (W8) that was previously suggested to play a key role in stabilization of the inner gate[36]. The dramatically lowered affinity for L-Leu upon mutation of W8 (W8A) supports that the mutant assumes a more inward-facing configuration with an unfavorable conformation of the binding pocket for L-Leu. Conversely, the preserved affinity for L-Phe suggests that this more inward-facing configuration is relatively more favorable for accommodating L-Phe. Interestingly, LeuT$_{WT}$ has been crystallized in the presence of the aromatic amino acid analog L-4-F-Phe in the outward-facing occluded state. The structure revealed a strained configuration consistent with a less favorable conformation of the binding pocket in the outward-facing occluded state for aromatic amino acids as compared to amino acids like L-Leu[39]. The uptake properties also support a major change in

conformation of W8A with abrogated transport of L-[$^3$H]Ala, indicating that release of the N-terminus and switching of the transporter towards a more inward-facing configuration impairs the ability of L-Ala to promote transport. We have no immediate explanation for the very low, yet detectable transport that we observe for L-[$^3$H]Phe in W8A other than it reflects the relatively better interaction of L-Phe with the inward-facing W8A conformation. Of further interest, a more general role of this tryptophan among NSS proteins has emerged from mutation of the equivalent residue in hDAT (W63), where a shift of the transporter towards an inward-oriented configuration was evidenced by increased constitutive internalization of the mutant forms[43]. Moreover, our MD simulations of the outward-facing occluded state showed that substitution of W8 in LeuT with Ala facilitates conformational transitions with dissociation of inner gates and water penetration through a channel to the Na2 site. Given the key role of these dynamic processes in the conformational isomerization of NSS proteins from the occluded to inward-facing states, our MD simulations of W8A were able to sample the initial stages of such structural transformation. Indeed, transitions from the occluded to the intermediate state have been observed to follow the similar initial dynamics and to involve slower modes of motions such as partial unwinding of TM5 segment and destabilization of the Na2, as observed in the large set of MD simulations of the mammalian DAT[20]. However, it is important to note that even in these systems that exhibit much faster dynamics at ambient temperatures than the prokaryotic LeuT and contain additional long termini segments that significantly contribute to the functional dynamics[44], these complete transitions were rare events.

We crystallized the Na$^+$/L-Phe-bound LeuT$_{W8A}$ and determined the structure at 2.6 Å resolution. In comparison to the outward-facing occluded state, the structure shows how the W8A mutation leads to release of the N-terminus and opening of the transporter to the inside, which occurs concomitantly with a closure of the extracellular vestibule. Comparison of the LeuT$_{W8A}$ structure with both MhsT structures[13] and with the apo inward-facing open state of LeuT[11] reveals how the structures might sequentially describe inward opening of the transporter with the N-terminus and cytoplasmic end of TM5 as central components (Fig. 7). In MhsT$^{HILIDE}$, the N-terminus stabilizes an unwound cytoplasmic end of TM5, whereas in MhsT$^{LCP}$ the N-terminal segment is disordered and TM5 is reverted to a continuous, but slightly kinked helix[13]. In the LeuT$_{W8A}$ structure, the N-terminus is also disordered and, moreover, the cytoplasmic end of TM5 has moved substantially outward relative to the helical bundle to assume an orientation similar to that seen in the apo inward-facing open state of LeuT (Figs. 5 and 6)[11]. In further support of a key role of TM5 in the transport process, the recent ibogaine-bound hSERT cryo-EM structures revealed both a movement of the cytoplasmic end of TM5 towards the membrane and helix unwinding in the apo inward-facing open configuration of the transporter[17]. Also, functional studies have pointed to the importance of conformational rearrangements of TM5. As an example, the application of the substituted cysteine accessibility method to SERT pointed towards substrate-induced architectural changes at the cytoplasmic end of this helix[7,45] and our previous use of SDFQS substantiated substrate-driven TM5 movements in LeuT[29].

In comparison to the MhsT structures and our LeuT$_{W8A}$ structure, TM1 adopts an orientation in the apo inward-facing open state of LeuT with an outward 45° tilt of TM1a away from the scaffold domain (Fig. 5b)[11]. Thus, TM1 displays the largest divergence between the LeuT$_{W8A}$ structure and the completely open transport intermediate (Fig. 5d, e). The magnitude of the TM1a movement has been challenged by intramolecular distance measurements using lanthanide-based resonance energy transfer

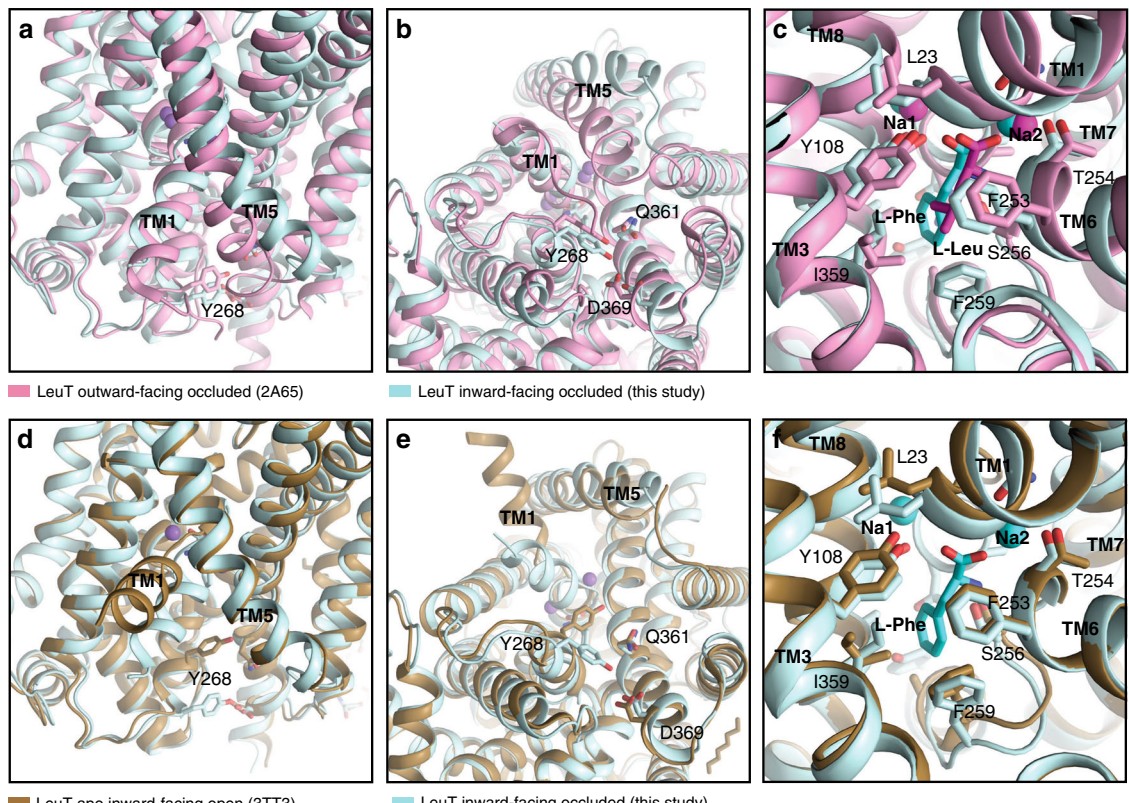

**Fig. 5 Comparison of the LeuT$_{W8A}$ structure with LeuT in the outward-facing occluded state and the apo inward-facing open state. a–c** Overlay of the LeuT$_{W8A}$ structure in inward-facing occluded conformation (this study; light blue) with LeuT in the outward-facing occluded state (PDB-ID: 2A65[10]; light pink). **a** Close-up view of the intracellular vestibule region from the membranous side with transmembrane segments (TMs) 1, 5, and Y268 residue labelled. **b** Close-up view of the intracellular vestibule from the cytoplasmic side with TM1, 5 and internal gate residues (Y268, Q361, and D369) labelled and shown as colored sticks. **c** Close-up view of sodium and substrate binding sites with selected coordinating residues labelled and shown as colored sticks. For each structure, two sodium ions are shown as spheres and substrates as darker sticks in the corresponding colors. Residues are selected within a distance of 4 Å. **d–f** Overlay of the LeuT$_{W8A}$ structure in inward-facing occluded conformation (this study; light blue) with LeuT structure in apo inward-facing open state (PDB-ID: 3TT1[11]; light brown). Panel presentation is identical as in **a–c**.

(LRET)[34]. However, in the inward-facing open conformation of the ibogaine-bound hSERT cryo-EM structure, the cytoplasmic end of TM1 (TM1a) is tilted outward almost to a similar degree (40°)[17]. This strongly suggests that the tilt of TM1 is required for release. Limited water access to the Na2 site was previously seen in both MhsT structures[13], but the passage appears insufficient for release, as the substrate and the two sodium ions remain bound. Furthermore, the cytoplasmic cavity in the LeuT$_{W8A}$ structure is larger due to the displaced N-terminus and shifted ICL2, and the Na2 site is solvent accessible through a wider penetration pathway, although with minor influence on the position of sodium ions and substrate. These considerations point further towards a critical functional role of TM1 movement away from the helical bundle for release of sodium and substrate (as demonstrated by the open structure). They also suggest that the conformational changes of TM5/ICL2 observed here allow release of the N-terminus prior to the penultimate shift of TM1, but not substrate release in the absence of the shift of TM1.

The interaction network of the sodium ions and substrate is remarkably conserved in all previously determined LeuT structures. The sodium ion in Na1 is octahedrally coordinated and interacts directly with the substrate, whereas a trigonal bipyramidal geometry characterizes the binding of sodium ion in Na2. In the MhsT structures[13], both bound sodium ions display octahedral coordination with an added water molecule to the Na2 site from the cytoplasmic environment, but no such water molecules are observed in LeuT structures[41]. Assessment of the

details of the coordination network of the sodium ions in the LeuT$_{W8A}$ structure, however, reveals subtle, but interesting differences compared to the previous structures of LeuT. In the present structure, both sodium ions shift towards the cytoplasm in accordance with the initial steps of substrate release. Notably, the location of the sodium sites is similar to those observed in the MhsT structures (Fig. 6), but the maintained coordination chemistry in the LeuT structures (preceding and succeeding the transport cycle state structurally determined for MhsT) may suggest that the overall principles rather than details of the release mechanisms are conserved.

Considering our findings of how changes in both TM5 and the N-terminus support inward opening, a mechanistic model might be suggested in context of the structures available for LeuT, MhsT, and hSERT. We propose that the dynamics between outward- and inward-facing occluded states explore the well-known structure of the outward-facing occluded state of LeuT[10] and the inward-facing occluded state with an unwound TM5[13]. Importantly, for any visit to the inward-facing occluded state, water will reach the Na2 site. This will in turn stimulate the dynamics of the N-terminus that may detach from the W8 and R5 anchor points and dissociate from the core. Once the N-terminal tail dissociates, the unwound TM5 may again adopt a helical structure and move outward as seen in the present structure, permitting further dynamics and solvation at the Na2 site. This is then followed by outward tilting of TM1a, which allows the full release of substrate and ions. In the inward-open

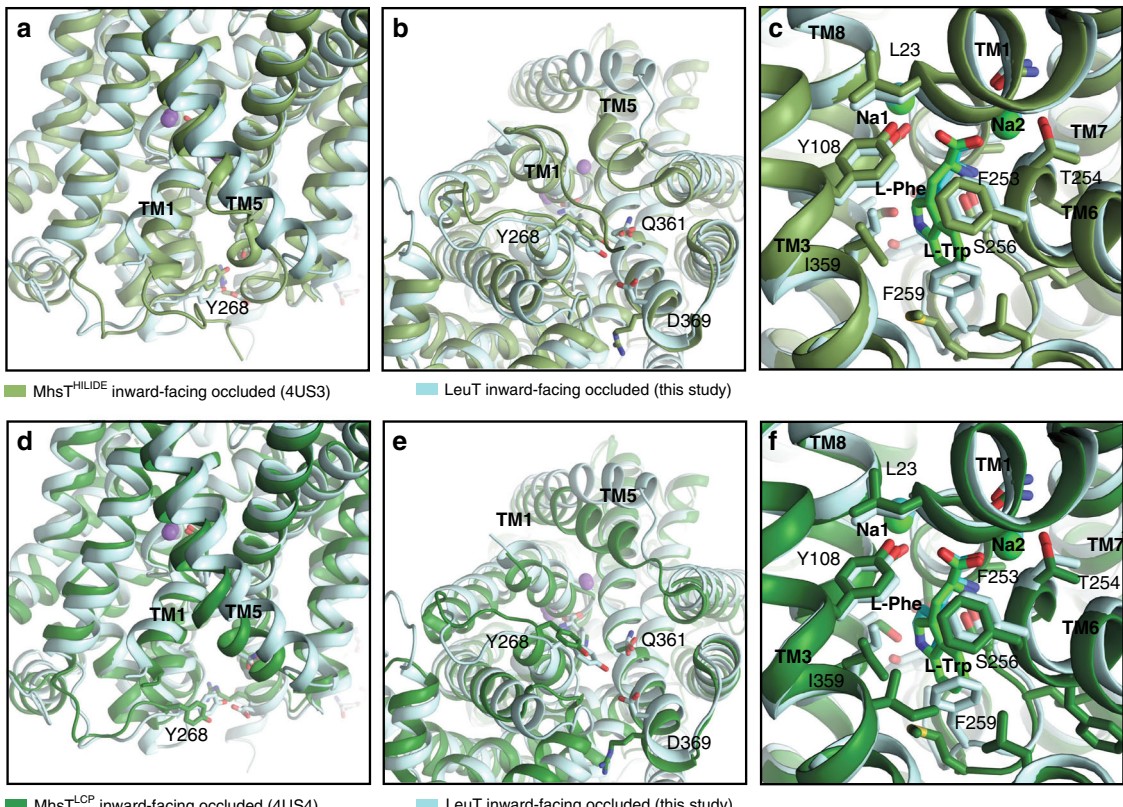

**Fig. 6 Comparison of the LeuT_{W8A} structure with the structures of MhsT. a–c** Overlay of the LeuT_{W8A} structure in inward-facing occluded conformation (this study; light blue) with the MhsT^{HILIDE} structure in inward-facing occluded state (PDB-ID: 4US3[13]; light green). **a** Close-up view of the intracellular vestibule region from the membranous side with transmembrane segments (TMs) 1, 5, and Y268 residue labelled. **b** Close-up view of the intracellular vestibule from the cytoplasmic side with TM1, 5 and internal gate residues (Y268, Q361, and D369) labelled and shown as colored sticks. **c** Close-up view of sodium and substrate binding sites with selected coordinating residues labelled and shown as colored sticks. For each structure, two sodium ions are shown as spheres and substrates as darker sticks in the corresponding colors. Residues are selected within a distance of 4 Å. **d–f** Overlay of the LeuT_{W8A} structure in inward-facing occluded conformation (this study; light blue) with the MhsT^{LCP} structure in inward-facing occluded state (PDB-ID: 4US4[13]; dark green). Panel presentation is identical as in **a–c**.

hSERT cryo-EM structure TM5 remains unwound, while at the same time TM1 moves outward[17]. TM5, however, is not unwound in the equivalent apo inward-facing open state of LeuT[11] and recent hydrogen-deuterium-exchange mass spectrometry (HDXMS) data on hSERT and dDAT do not indicate helical unwinding of TM5[46,47]. On the other hand, HDXMS studies on LeuT suggested helix unwinding on the cytoplasmic side (including TM5), presumably occurring as part of a conformational shift towards an inward-oriented state in detergent environment[32]. Indeed, such unwinding can likely also occur in a lipid membrane environment as supported by similar HDXMS findings on LeuT in detergent micelles and upon reconstituting the transporter into lipid bilayer nanodiscs[32]. Thus, further studies are needed to fully clarify the role of helix unwinding in the NSS translocation process.

Summarized, by determining the structure of an important intermediate of LeuT, the present study adds critical insight into the LeuT transport cycle that now is represented by five distinct conformational intermediates (Fig. 7). A similar series of high-resolution states is not available for any other NSS member, which underlines the overall importance of LeuT for our current understanding of the ion-coupled translocation mechanism for this important family of membrane transporters. Indeed, the available structures provide an essential framework for further untangling delicate mechanistic aspects of the physiological and pathophysiological functions of NSS proteins as well as for

permitting more efficient pharmacological targeting to combat the many diseases where the NSS proteins play an important role.

## Methods

**Single-molecule FRET imaging experiments.** LeuT variants were expressed in E. coli C41 (DE3; commercial strain available from Lucigen Corporation, USA) using pQO18-TEV vector derivatives with the mutations H7C/R86C (intracellular)[26]. Protein solubilized in n-dodecyl-β-D-maltopyranoside (DDM; Anatrace, USA) was purified using immobilized metal affinity chromatography (IMAC) on Ni^{2+} Sepharose 6 FastFlow column (GE Healthcare, USA). LeuT was labeled with maleimide-activated LD550 and LD650 (Lumidyne Technologies, USA) at a 1:1.5 molar ratio (200 μM total) for 1 h at 4 °C, followed by size exclusion chromatography (SEC) using a Superdex 200 16/60 column (GE Healthcare, USA). All experiments were performed at 25 °C in buffer containing 50 mM Tris/MES pH 7.0, 5 mM NaCl, 195 mM KCl, 10% glycerol, 0.05% (w v^{−1}) DDM and 1 mM 2-mercaptoethanol. An oxygen-scavenging environment consisting of 0.2 units per mL purified glucose oxidase, 1.8 units per μL purified catalase (both from Sigma-Aldrich, USA) and 0.1% (v v^{−1}) glucose was used in all experiments to minimize photobleaching.

Fluorescence experiments were performed using a prism-based total internal reflection fluorescence (TIRF) microscope[28,48,49]. Microfluidic imaging chambers passivated with a mixture of PEG and biotin-PEG were prepared with 0.8 μM streptavidin (Invitrogen, USA) and 4 nM biotin-tris-(NTA-Ni^{2+})[50]. Fluorophore labeled, His-tagged LeuT molecules were reversibly immobilized to the surface-associated Ni^{2+} atoms. LD550 fluorophores were excited by the evanescent wave generated by total internal reflection (TIR) of a 532 nm diode-pumped solid-state laser (Laser Quantum, USA). Photons emitted from LD550 and LD650 were collected using a 1.27 NA ×60 PlanApo water-immersion objective (Nikon, USA) and a MultiCam-LS device (Cairn, USA) with a T635lpxr-UF2 dichroic mirror (Chroma, USA) to project the two spectral channels onto synchronized sCMOS cameras (Flash 4.0 v2, Hamamatsu, USA). Fluorescence data were acquired using

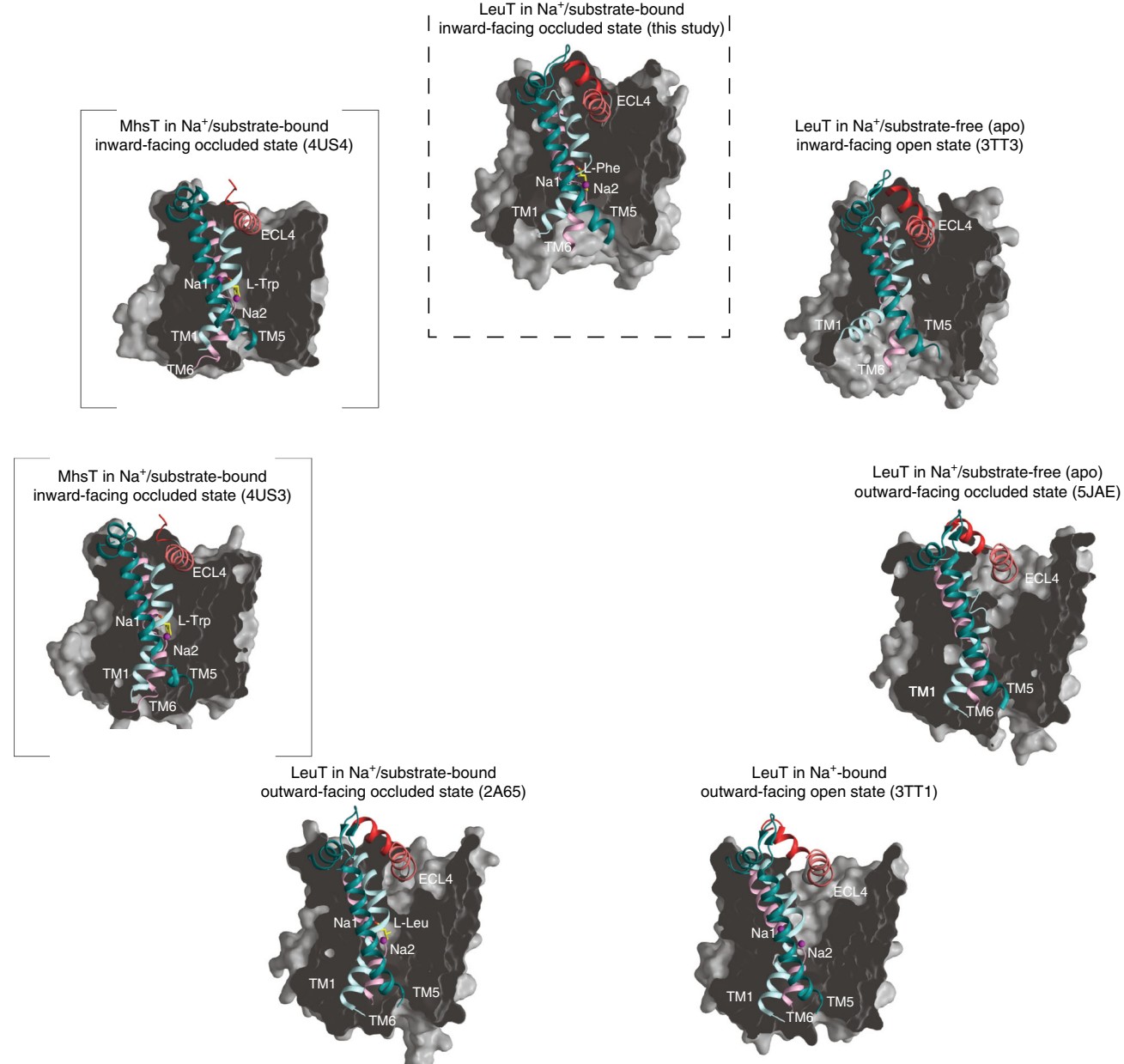

**Fig. 7 Structures of LeuT and MhsT organized clockwise after progressive opening to the inside.** The structures are shown as cross-sectional representations organized clockwise after progressive opening to the inside starting with the $Na^+$/substrate free outward-facing occluded LeuT state (PDB-ID: 5JAE[12]; middle right) followed by the $Na^+$-bound outward-facing open LeuT state (PDB-ID: 3TT1[11]; bottom right), the $Na^+$/L-Leu-bound outward-facing occluded LeuT state (PDB-ID: 2A65[10]; bottom left), the $Na^+$/L-Trp-bound inward-facing occluded MhsT state with an unwound TM5 (PDB-IDs: 4US3[13]; middle left and in square brackets), the $Na^+$/L-Trp-bound inward-facing occluded MhsT state with helical TM5 (PDB-IDs: 4US4[13]; upper left and in square brackets), the $Na^+$/L-Phe-bound inward-facing occluded state of LeuT (this study; top middle) and the apo inward-facing open LeuT state (PDB-ID: 3TT3; top right). For clarity, only the orientation of transmembrane segments (TMs) 1, 5, and 6, and extracellular loop (ECL) 4 is shown to illustrate the most significant concerted movements related to opening and closure of the intracellular and extracellular vestibules, respectively. Sodium ions and substrate molecules are shown as purple spheres and yellow sticks, respectively. The structures support a model in which $Na^+$/substrate reach the outward-facing binding sites via open extracellular vestibule, leading to its closure by the conformational rearrangements involving TM1, 5 and ECL4. The adapted inward-facing occluded conformation transits subsequently towards the inward-facing state, largely by the release of the N-terminal tail, disruption of the interactions provided by internal gate residues and changes in TM5 dynamics. Specifically, the structures illustrate how changes in TM5 dynamics involving putative cytosolic unwinding followed by a progressive outward movement towards the membrane (from PDB-ID: 2A65 over PDB-ID: 4US3 and PDB-ID: 4US4 to the present structure of LeuT$_{W8A}$) putatively lead to formation of a large cytoplasmic cavity allowing access to Na2. Furthermore, as the transport cycle progresses, TM1 swings away from TM5 and 6 to open the intracellular vestibule that permits the release of bound $Na^+$/substrate (PDB-ID: 3TT3). Finally, following $Na^+$/substrate diffusion, TM1 and 5 pull back, and ECL4 uncaps the extracellular vestibule, facilitating the return to the apo outward-facing open state able to bind $Na^+$/substrate again.

custom software implemented in LabView (National Instruments, USA) at 40 frames per second (25 ms time resolution).

Analysis of single-molecule fluorescence data was performed using SPARTAN version 3.7[48]. Spectral bleed-through was corrected by subtracting a set fraction (0.165) of the donor intensity from the acceptor channel. FRET traces were calculated as $E_{FRET} = I_A/(I_A + I_D)$, where $I_A$ and $I_D$ are the donor and acceptor fluorescence traces, and set to zero whenever the donor was in the dark state. Traces were selected for further analysis using the following criteria: (i) single-step donor photobleaching; (ii) $SNR_{Background} \geq 8$; (iii) $SNR_{Signal} \geq 5$; (iv) less than four donor blinking events; and (v) FRET efficiency above 0.15 for at least 300 frames (7.5 s). smFRET traces were then idealized by the segmental K-means algorithm[51], using a 4-state model with FRET values of 0, 0.47, 0.63, and 0.79. Kinetic parameters were estimated using the Maximum Interval Likelihood algorithm[52] implemented in SPARTAN.

**Molecular constructs for computational experiments.** The crystal structure of WT LeuT (LeuT$_{WT}$; PDB-ID: 3GJD) was used to create an atomistic full-length model of the LeuT$_{W8A}$ mutant[53]. This model represents LeuT in the occluded state, with L-Leu substrate bound in the primary binding S1 site and with both Na1 and Na2 sites occupied by Na$^+$ ions (termed here Na$^+$/Na1 and Na$^+$/Na2 ions, respectively). The molecule of $n$-octyl-β-D-glucopyranoside (βOG) detergent overlying with the putative secondary binding site (S2) was removed. LeuT residues missing in the crystal structure (i.e., first 4 residues of the N-terminus, last 8 residues of the C-terminus, and the P132-N133-A134 stretch in the ECL2) were added with Modeller[54]. W8A was introduced into full-length LeuT$_{WT}$ structure using the VMD mutator plugin[55]. In addition to the above-mentioned LeuT$_{W8A}$ model (hereafter referred to as W8A$_{OUT-OCC}$), other LeuT structures were prepared for simulation: (i) LeuT$_{WT}$ based on the 3GJD X-ray structure (termed here WT$_{OUT-OCC}$); and (ii) the crystal structures of the LeuT$_{W8A}$ mutant reported here with the two chains of the crystallographic unit cell separated (referred to as W8A$_{chainA}$ and W8A$_{chainB}$, respectively and containing residue segments 13–511 and 13–507 of LeuT in complex with L-Phe substrate in the S1 site and two sodium ions in the respective binding sites). The protonation states of the titratable residues in all the LeuT models were evaluated with the ProPka software[56], resulting in the protonation of E112, E287 and E419 in all molecular constructs[19,57]. The W8A$_{OUT-OCC}$ and WT$_{OUT-OCC}$ models were capped by the standard charged N- and C-termini, whereas W8A$_{chainA}$ and W8A$_{chainB}$ structures were capped with neutral termini (i.e., acetylated N-terminus and methylamide C-terminus). For the simulations, the W8A$_{OUT-OCC}$, WT$_{OUT-OCC}$, W8A$_{chainA}$ and W8A$_{chainB}$ LeuT constructs were inserted into a lipid bilayer composed of a 74:26 mixture of 1-palmitoyl-2-oleoyl-sn-glycero-3-phosphoethanolamine (POPE)/1-palmitoyl-2-oleoyl-sn-glycero-3-phosphoglycerol (POPG) in a total of 296 lipids, which also served in microsecond scale MD simulations of the LeuT[58]. The protein-membrane complexes were then surrounded by a water box containing TIP3P water and 0.15 M NaCl ionic concentration, for a total atom count of ~120.000 atoms in each construct.

**Equilibration protocol and force-field parameters.** Each of the systems was first subjected to a multi-step equilibration protocol performed with the NAMD software version 2.11 suite, and CHARMM36 parameters for proteins, lipids and ions[58]. Briefly, this phase included: (i) minimization for 5.000 steps and MD simulation with 1-fs integration time-step for 250 ps, fixing all atoms in the system except for the lipid tails; (ii) minimization for 2.500 steps and MD simulation with 1-fs time-step for 500 ps with constrained protein backbone and lipid headgroups (force constant of 1 kcal mol$^{-1}$ Å$^{-2}$), keeping water out of the membrane hydrophobic core; (iii) gradual release of the constraints on the protein backbone and lipid headgroup atoms (force constant of 0.5 and 0.1 kcal mol$^{-1}$ Å$^{-2}$), while still keeping water out of the membrane interior. At each value of the force constant, the system was minimized for 2.500 steps followed by 500 ps MD (with 1-fs time-step). (iv) consisted of unbiased MD simulation for 30 ns using a 2-fs time-step. These steps implemented PME for electrostatics interactions[59], and were carried out in the NPT ensemble under semi-isotropic pressure coupling conditions, at 310 K. Also, the Nosé-Hoover Langevin piston algorithm[60] was used to control the target $P = 1$ atm pressure with the LangevinPistonPeriod set to 100 fs and LangevinPistonDecay set to 50 fs. The van der Waals interactions were calculated applying a cutoff distance of 12 Å and switching the potential from 10 Å.

**Ensemble MD simulations.** For all the molecular LeuT constructs, WT$_{OUT-OCC}$, W8A$_{OUT-OCC}$, W8A$_{chainA}$ and W8A$_{chainB}$, the equilibration phase was followed by ensemble MD simulations during which each system was simulated in 10 statistically independent replicates using unbiased MD. For the WT$_{OUT-OCC}$ model, ~10 µs total simulation time was accumulated (1 µs per replicate), and the W8A$_{OUT-OCC}$, W8A$_{chainA}$, and W8A$_{chainB}$ systems were simulated for ~7 µs (700 ns per replicate). These ensemble simulations were carried out using ACEMD software developed at Acellera[61] and with CHARMM36 force-field parameters. The simulations with ACEMD implemented the PME method for electrostatic calculations[20,21,58,61], which included 4-fs integration time-step enabled by standard mass repartitioning procedure for hydrogen atoms. The computations were

conducted under the NVT ensemble (at $T = 310$ K), using the Langevin Thermostat with Langevin Damping Factor set to 0.1.

**Quantification of hydration in MD simulations.** To quantify water penetration through the intracellular vestibule of LeuT in the MD trajectories we used protocols developed for the homologous hDAT[20]. Here a water molecule is considered to be inside the intracellular vestibule of the protein if its oxygen atom is within 15 Å of the center of mass of the bound substrate, but not within 5 Å of any lipid, and if its z-coordinate lies between z-coordinates of C$_\beta$ atom of F259 and C$_\alpha$ atom of G13. In addition, the following criteria are enforced: the water oxygen atom should not be: (i) within 8 Å of the backbone of L495; (ii) within 5 Å of the backbone of L257; and (iii) within 13 Å of the backbone residues V276, L280 and L420. We focused specifically on the channel reaching from the intracellular end of the transporter to the substrate binding region, which is defined as the region in the intracellular vestibule that is within 12 Å of the backbone of residue A198.

**Protein expression and purification.** The cDNA encoding the LeuT$_{WT}$ from Aquifex aeolicus (UniProt accession number O67854) C-terminally fused with thrombin cleavage site and 8xHis-tag was cloned into the pET16b vector. LeuT$_{W8A}$ variant was generated using QuikChange site-directed mutagenesis kit (Agilent Technologies, USA) with the following primers: 5′–GTTAAAAGGGAACACGCG GCGACGCGACTCGG–3′ and 5′– CCGAGTCGCGTCGCCGCGTGTTCCCTTT TAAC–3′. LeuT$_{WT}$ and W8A mutant were expressed in E. coli C41 (DE3; commercial strain available from Lucigen Corporation, USA)[30]. Isolated crude bacterial membranes were solubilized in 1% DDM (Anatrace) and the protein was purified using IMAC employing ProBond Ni-IDA resin (Thermo Fisher Scientific, USA). For functional studies, bound protein was eluted in IMAC buffer composed of 20 mM Tris pH 8.0, 200 mM KCl, 20% (v v$^{-1}$) glycerol, 0.05% (w v$^{-1}$) DDM and 300 mM imidazole. For crystallization studies, LeuT$_{W8A}$ mutant was purified in the following IMAC buffer: 20 mM Tris pH 8.0, 200 mM NaCl, 100 µM L-Phe, 20% (v v$^{-1}$) glycerol, 0.05% (w v$^{-1}$) DDM and 300 mM imidazole. Subsequently, IMAC-pure protein was concentrated on Vivaspin 20 column (MWCO 50 kDa; Sartorius, Germany) and subjected SEC using Superdex increase 200 10/300 GL column (GE Healthcare) equilibrated in SEC buffer composed of 20 mM Tris pH 8.0, 200 mM NaCl, 1 mM L-Phe and 1% (w v$^{-1}$) βOG (Anatrace).

**Uptake and binding experiments.** For uptake experiments, purified LeuT variants were reconstituted at a 1:150 [w w$^{-1}$] ratio into preformed liposomes (composed of total E. coli lipids/1-palmitoyl-2-oleoyl-glycero-3-phosphocholine (POPC) at a 3:1 [w w$^{-1}$] ratio) in 100 mM potassium phosphate pH 6.5, 2 mM β-mercaptoethanol. Uptake of L-[$^3$H]Ala or L-[$^3$H]Phe (both were from American Radiolabeled Chemicals, Inc., USA and used at a specific activity of 10 Ci mmol$^{-1}$) in proteoliposomes containing WT LeuT or W8A LeuT, or in control liposomes lacking LeuT, was measured in the presence of 10 mM Tris/MES pH 8.5, 150 mM NaCl for the indicated time periods. Samples were quenched by the addition of ice-cold 100 mM potassium phosphate pH 6.0, 100 mM LiCl and filtered through 0.45 µm nitro-cellulose filters (Millipore, USA). The accumulated radioactivity on the filters was detected with scintillation counting (Hidex 300SL liquid scintillation counter; Hidex, Finland) and the dpm were translated in pmol using known standards of the respective radiolabeled compounds. Data (means ± SEM of two independent experiments with technical triplicates) were normalized to the amount of LeuT used in the uptake experiments. The concentration of LeuT in the proteoliposomes was quantified from silver-stained SDS-polyacrylamide gels by densitometry with the ImageJ software (NIH, USA) using known amounts of purified LeuT as standard.

Binding experiments were performed using the scintillation proximity assay (SPA)[62]. Saturation binding was determined for 1 µg mL$^{-1}$ of purified protein in a clear-bottom 96-well plate, with 1.25 mg mL$^{-1}$ of YSi-Cu His-tag SPA beads (PerkinElmer, USA) in binding SPA buffer A (20 mM Tris-HCl pH 8.0, 200 mM NaCl, 20% (v v$^{-1}$) glycerol and 0.05% (w/v) DDM) in the presence of 0.5–10000 nM [$^3$H]-Leu (5.02 Ci mmol$-1$) or 5–4000 nM [$^3$H]-Phe (2.5 Ci mmol$^{-1}$; both from PerkinElmer). Nonspecific background was determined in the presence of 10 mM non-radioactive L-Leu. Plates were sealed and incubated for ~16 h at 4 °C shielded from light, and activity was monitored using 2450 MicroBeta$^2$ microplate counter (PerkinElmer). Data were fitted to a non-linear regression analysis (one site-specific binding or sigmoidal dose-response) using GraphPad Prism 6.0 (GraphPad Software, USA).

**Crystallization and structure determination.** LeuT$_{W8A}$ crystals were grown by hanging-drop vapor diffusion at 18 °C. Briefly, SEC-grade protein sample (~4 mg mL$^{-1}$) was mixed with a reservoir solution composed of 100 mM HEPES-NaOH pH 7.2, 100 mM sodium formate, 14% (v v$^{-1}$) PEG 3350 and 15% (v v$^{-1}$) glycerol (all chemicals from Sigma-Aldrich). Crystals were mounted in cryo-loops directly from the mother liquor and flash frozen in liquid nitrogen. X-ray diffraction data were collected at beamline X06SA (Swiss Light Source, SLS) at a wavelength of 0.9999 Å and at 100 K. Multiple datasets were collected on a single large crystal. Processing of the data was done using XDS[63]. BLEND[64] was used for examining data clusters and combining datasets, and AIMLESS[65] was used for data reduction and scaling (Table 1). Phasing of LeuT$_{W8A}$ was done using PHENIX.MR[66] and

PHASER[67]. To identify the best search, model screening using all available LeuT structures available in the PDB at the time was performed. The apo inward-facing open LeuT structure (PDB-ID: 3TT3) showed the best Z-score and LLG, and it was used for phasing in a truncated form starting from residue 28. Iterative manual rebuilding of the model was performed in COOT[68] and the refinement with PHENIX.REFINE[66]. Before the final round of refinement, the model was uploaded to the NAMDINATOR server (http://namdinator.au.dk)[69] for an automated MDFF simulation combined with PHENIX real space refinement and the resulting model was subsequently subjected to a final refinement round with PHENIX. REFINE (Table 1). Ramachandran statistics showed 97.67% in favored, 2.33% in allowed and no outlier residues. Figures were generated with PyMOL (http://www.pymol.org). Electrostatic potential energy maps were built employing APBS PyMOL plugin[70]. Movies were created using the MORPHINATOR (http://morphinator.au.dk) tool developed by Jesper L. Karlsen, Aarhus University. In MORPHINATOR morphs between the outward-open LeuT structure (PDB-ID: 3TT1), the outward-occluded LeuT structure (PDB-ID: 2A65), the inward-occluded LeuT structure (this study), the inward-open apo LeuT structure (PDB-ID: 3TT3) and the outward-return apo LeuT structure (PDB-ID: 5JAE) were generated in an automated manner by employing superpose_pdbs and geome-try_minimization from the PHENIX software suite. The 3TT1, 3TT3, and 5JAE structures were mutated to the WT sequence and the loop between TM3 and EL2 was modelled in the 3TT1, 2A65, and 5JAE structures based on the TM3-EL2 loop in the inward-occluded structure from this study in order to make an unbroken morph.

**Reporting summary**. Further information on research design is available in the Nature Research Reporting Summary linked to this article.

## Data availability

Data supporting the findings of this paper are available from the corresponding authors upon request. A reporting summary for this Article is available as a Supplementary Information file. The source data underlying Figs. 1 and 3 are provided as a Source Data file.

Coordinates and structure factors have been deposited in the Protein Data Bank under accession code 6XWM.

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

## Acknowledgements

K.G. and P.G. were supported by the Lundbeck Foundation Fellowship R133-A12689. P.G. was supported by the Knut and Alice Wallenberg Academy Fellow (KAW2015.0131). The work was supported by NIH Grants P01 DA012408 (U.G., J.J. and H.W.) and R01 DA041510 (J.A.J. and H.W.), the UNIK Center for Synthetic Biology (U.G.), Lundbeck Foundation Center for Biomembranes in Nanomedicine (2009-4585) (U.G.), the Independent Research Fund Denmark (Sapere Aude 0602-02100B, FP1 0602-02100B and 4183-00581) C.J.L.), the European Community's Seventh Framework Programme FP7/2007-2013 HEALTH-F4-2007-201924 (U.G.) and the Carlsberg Foundation (CF17-0171) (C.J.L.). T.B. and P.N. were supported by Lundbeck Foundation grant no. DANDRITE-R248-2016-2518. The following computational resources are gratefully acknowledged: resources of the Oak Ridge Leadership Computing Facility (INCITE allocation BIP109) at the Oak Ridge National Laboratory, which is supported by the Office of Science of the U.S. Department of Energy under Contract No. DE-AC05-00OR22725; an allocation at the National Energy Research Scientific Computing Center (NERSC, repository m1710) supported by the Office of Science of the U.S. Department of Energy under Contract No. DE-AC02-05CH11231; and the computational resources of the David A. Cofrin Center for Biomedical Information in the HRH Prince Alwaleed Bin Talal Bin Abdulaziz Alsaud Institute for Computational Biomedicine at Weill Cornell Medical College. We thank Jesper L. Karlsen, Aarhus University, for assistance with numerous aspects of scientific computing, and we thank Lina Malinauskaite (MRC Laboratory of Molecular Biology, Cambridge) for input to Fig. 7. We are also grateful for assistance with crystal screening and data collection at the Swiss Light Source, the Paul Scherrer Institute, Villigen, beam line X06SA. Access to synchrotron sources was granted by The Danscatt program of the Independent Research Fund Denmark.

## Author contributions

K.G. and J.S.M. performed LeuT$_{W8A}$ protein expression and purification. K.G. crystallized LeuT$_{W8A}$. K.G. and T.B. collected X-ray diffraction data. T.B. solved and refined the crystal structure. J.S.M. performed binding experiments and analyzed the results together with C.J.L. G.K. and H.W. conducted MD simulations and with M.V.L. analyzed the results. D.T., S.C.B, and J.A.J performed the smFRET experiments and interpreted the results. M.Q. and J.A.J. performed the uptake experiments and analyzed the results. K.G., T.B., P.G., P.N., and U.G. analyzed structural data. K.G., J.S.M., C.J.L., P.N., and U.G. designed the project. K.G., T.B., and J.W.M. generated figures. K.G., T.B. P.N., and U.G. wrote the paper with contribution from all authors.

## Competing interests

The authors declare no competing interests.
