## [Peer Review File · Nature Communications]

Reviewers' Comments:

Reviewer #1:

Remarks to the Author:

LeuT has become an important structural model for the NSS family. Human members of this family are important drug targets, and improved understanding of substrate binding, transport, and release provide potential insights into the molecular-level functional details of these biologically and biomedically important transporters. The manuscript "Mechanism of substrate release in neurotransmitter:sodium symporters: the structure of LeuT in an inward-facing occluded conformation" by Gotfryd, et al is valuable because it does not simply solve another LeuT structure, it includes single molecule FRET and MD simulations to develop the strategy used to trap the inward-occluded state and more rigorously connect the structure to a particular functional state. The importance of this model system and the insight on substrate and ion release provided by this intermediates state structure make this manuscript of interest to the broad audience of Nature Communications. The manuscript is clear and well organized. There are a few, generally minor flaws that should be addressed before the manuscript is suitable for publication.

1. In the beginning of the results section, the authors discuss the effect of Phe binding on both the population of an intermediate FRET state and the dynamics of the transporter under these conditions. This is near clearly resolved. Does Phe stabilize an intermediate or increase dynamics/rates of transition to and from this intermediate state?

2. It is well established that comparison of structures of different homologs in an attempt to build a more complete structural understanding of the transport cycle can be misleading or incorrect since different homologs will have additional structural differences. Thus, Fig. 7 is the weakest aspect of the manuscript. Illustrating only the differences within the LeuT structures of different mutants/different substrates/no substrate would be less potential to be misleading or confusing and still illustrate the characterization of where this particular structure falls within the transport cycle. In Figures 7, the images are also quite dark making it difficult to understand the figure. The highlighting of accessible surface isn't particularly clear in illustrating the differences between some of the conformations – such as inward open vs inward facing occluded (clear change in N-term of TM1, but not at all clear how this change affects whether the binding pocket is accessible to the aqueous environment or not in this orientation. Also, it would be helpful to have TM6 drawn as well, for comparison with Fig 1 and earlier structure figures that highlight the relative motion of TM1/5/6 in the opening process.

3. Only one figure shows the electron density, difficult to judge the quality of key portions of the structure. It would be helpful to have the electron density maps shown for the Phe/sodium binding sites as well.

4. In the figures illustrating the crystal structure and differences with Leu-bound outward-facing occluded state:

-Fig 4b - It is helpful to have the overall structure presented in an orientation that allows the reader to understand the positioning within the membrane in at least one of the figures. Fig 4b appears to be at an odd angle. Given the close-ups in a and c, it would be helpful to have figure 4b in a clean orientation (side on with rough membrane placement indicated). It is difficult to judge whether the structure is open or occluded when it isn't clear where the membrane and aqueous interfaces are.

-Fig. 4a, it is difficult to see the residues coordinating the two Na ions. I would prefer to see two sub figures, one for each Na, so that all of the coordinating side chains are visible.

-Figure S5 – having the residues from 2A65 hanging in space makes it difficult for a reader less familiar with these structures to orient and understand the consequences of slight changes in coordination geometry. Please include the backbone ribbon in gray for both structures so that the coupled changes in side chain orientation to coordinate the ions and TM helix orientation.

-Minor quibble - Fig. 5b, e are very faint. Reducing the depth cue slightly would help make the

figure more visible.

Reviewer #2:

Remarks to the Author:

The manuscript under review presents a new LeuT (W8A mutant) crystal structure in a Na⁺ and L-Phe bound inward facing occluded state. The choice of mutation and bound substrate in the crystallization process is guided by smFRET and MD data indicating that the W8A substitution and L-Phe binding stabilize inward facing conformations of LeuT. In the new crystal structure important interactions between gating residues in the N-terminus and TMs 1 and 8 are altered, leading to opening of the protein toward the cytoplasm due to movement of TM5 and potentially increased hydration (as evidenced from MD simulations) of the Na² binding site, necessary for future Na² dissociation. Comparison to existing outward facing occluded and inward facing opened structures of LeuT, as well as 2 available inward facing occluded structures of MhsT, show that the new LeuT structure features significant changes in the position of TM5 without much change in TM1, which places it as a novel intermediate structure between the outward facing occluded and inward facing open conformations in the catalytic cycle of LeuT. The study is well executed, with appropriate methodology and the conclusions are consistent with the provided results. Good understanding of the LeuT catalytic cycle is crucial for conceptual studies on secondary transport and may have immediate impact on pharmacological approaches with implications to mental health and opioid addiction. As such, the study is both novel and significant and is fit for publication in Nature Communications after a minor revision.

The authors report MD simulations on WT and W8A LeuT models based on an existing outward facing structure (3GJD), which guide their crystallization strategy. Later, MD simulations with the crystalized LeuT (W8A, Na/L-Phe bound) monomers are also reported, where water permeation toward the Na² site is visible. How do the structures from these two sets of calculations compare? Are there any indications of TM5 movement caused by the W8A mutation or substrate binding in the MD simulations with the 3GJD models (WT out-occ and W8A out-occ), consistent with the new crystallized structure? Considering the length of the simulations (1 microsecond per replica) and the number of replicas (10+10) even slow TM movements might be observable in some of the replicas of these ensembles. Did the hydration patterns at the end of the 1 microsecond in the 3GJD models resemble those of the new crystal structure? Did Na⁺ from Na² dissociate in any of the MD simulations? Basically, do the simulations, starting from the outward-facing occluded models lend support to the transport hypothesis of the authors (movement of TM5, no movement of TM1, increased hydration of Na² site) and that the new crystal structure is an intermediate in the catalytic cycle and not an artifact of the W8A mutation? A few words about this in the text of the manuscript will strengthen the drawn conclusions.

Overall, very interesting and thought-provoking studies that deserves publication in Nature Communication pending minor revisions noted above.

Reviewer #3:

Remarks to the Author:

Gotfryd et al. present the previously elusive state of LeuT in sodium/substrate bound inward occluded conformation at the relatively high resolution of 2.6 Å. This is a result of a nice and elegant approach of combination of smFRET and MD simulations that led to a proposal that this state can be arrested in presence of L-Phe and W8A mutant. I have no concerns about the structure - however looking at figure S4 - there are some FoFc peaks, if the sigma of 2fo-fc map is lowered is it possible to build extra residues? If so, the further calculation of the polder maps might be helpful. It would be also nice if the authors show omit maps for sodium ions and L-Phe. Apart from the structure I am puzzled with two observations - (1) the affinity to L-Phe is lower than to L-Leu despite that L-Phe has more favourable contacts - hydrophobic and aromatic and (2)

weird transport behaviour of WT and W8A mutant I think the authors should elaborate on that. The minor things: reaction cycle intermediates - I believe transport cycle intermediates is more appropriate. Please calculate the volume of enlarged cavity. Also maybe worth discussing whether such unwinding of helices (discussion) is feasible in membrane?

RESPONSE TO REVIEWERS (NCOMMS-16-00358A-Z)

We acknowledge the very careful evaluation of our manuscript and the constructive comments. As outlined below we have now addressed all comments from the Reviewers.

“Reviewer #1 (Remarks to the Author):

LeuT has become an important structural model for the NSS family. Human members of this family are important drug targets, and improved understanding of substrate binding, transport, and release provide potential insights into the molecular-level functional details of these biologically and biomedically important transporters. The manuscript “Mechanism of substrate release in neurotransmitter:sodium symporters: the structure of LeuT in an inward-facing occluded conformation” by Gotfryd, et al is valuable because it does not simply solve another LeuT structure, it includes single molecule FRET and MD simulations to develop the strategy used to trap the inward-occluded state and more rigorously connect the structure to a particular functional state. The importance of this model system and the insight on substrate and ion release provided by this intermediates state structure make this manuscript of interest to the broad audience of Nature Communications. The manuscript is clear and well organized. There are a few, generally minor flaws that should be addressed before the manuscript is suitable for publication.

We appreciate these favorable comments and are glad that the Reviewer finds the paper well suited for Nature Communications.

“1. In the beginning of the results section, the authors discuss the effect of Phe binding on both the population of an intermediate FRET state and the dynamics of the transporter under these conditions. This is near clearly resolved. Does Phe stabilize an intermediate or increase dynamics/rates of transition to and from this intermediate state?”

This is an interesting consideration. To address the question, we have made a major revision of Fig. 1 where we now also show an analysis of the kinetic parameters of LeuT intracellular dynamics. Since the dynamics in question were not completely resolved with the 100 ms time resolution previously employed, the experiments were repeated with 25 ms time resolution. As described in the updated manuscript, the dominant effects we observed in the presence of L-Phe at concentrations up to 1 μ M was an approximately 4-fold increase in the rate of transitioning from the low- to intermediate-FRET state and an approximately 2-fold increase in the rate of transitioning from the intermediate- to the low-FRET state. The net effect of this modulation of dynamic rates was to increase occupancy in the intermediate-FRET state and to increase the overall frequency of fluctuations, particularly between the low- and intermediate-FRET states. At higher concentrations of L-Phe, transitions between these states became so transient that they could not be resolved even with the faster experimental time resolution (25 ms), resulting in apparent stabilization of the intermediate-FRET state. We should note that we decided to take out data on L-Ala as these were just repetitive of previous experiments that are already published (see Refs. 26-28).

The description of the results (page 6, 1st paragraph – page 7, 1st paragraph) and the legend to Fig. 1 have been updated accordingly.

“2. It is well established that comparison of structures of different homologs in an attempt to build a more complete structural understanding of the transport cycle can be misleading or incorrect since different homologs will have additional structural differences. Thus, Fig. 7 is the weakest aspect of the manuscript. Illustrating only the differences within the LeuT structures of different mutants/different substrates/no substrate would be less potential to be misleading or confusing and still illustrate the characterization of where this particular structure falls within the transport cycle. In Figures 7, the images are also quite dark making it difficult to understand the figure. The highlighting of accessible surface isn’t particularly clear in illustrating the differences between some of the conformations – such as inward open vs inward facing occluded (clear change in N-term of TM1, but not at all clear how this change affects whether the binding pocket is accessible to the aqueous environment or not in this orientation. Also, it would be helpful to have TM6 drawn as well, for comparison with Fig 1 and earlier structure figures that highlight the relative motion of TM1/5/6 in the opening process. “

We understand the concern about Fig. 7 and agree that comparison between homologs can be misleading. Nevertheless, we still find that it is of interest to compare LeuT with the MhsT structures. We have therefore allowed ourselves to keep the overall organization of the figure in which the different structures are organized clockwise after progressive opening to the inside, as also specifically stated in the legend. However, we have now added brackets around the MhsT structures. It is also important to note that there are no arrows between the structures to underline that the figure does not describe the cycle *per se* but rather the known structures of prokaryotic NSS proteins. In the revised figure, we have also tried to make the illustrations of the transporter conformations lighter, and, as requested, we have included TM6. We hope that these changes comply with the Reviewer’s suggestions.

“3. Only one figure shows the electron density, difficult to judge the quality of key portions of the structure. It would be helpful to have the electron density maps shown for the Phe/sodium binding sites as well.”

These details have now been included in Supplementary Fig. 4.

“4. In the figures illustrating the crystal structure and differences with Leu-bound outward-facing occluded state:

-Fig 4b - It is helpful to have the overall structure presented in an orientation that allows the reader to understand the positioning within the membrane in at least one of the figures. Fig 4b appears to be at an odd angle. Given the close-ups in a and c, it would be helpful to have figure 4b in a clean orientation (side on with rough membrane placement indicated). It is difficult to judge whether the structure is open or occluded when it isn't clear where the membrane and aqueous interfaces are.

-Fig. 4a, it is difficult to see the residues coordinating the two Na ions. I would prefer to see two sub figures, one for each Na, so that all of the coordinating side chains are visible.

-Figure S5 – having the residues from 2A65 hanging in space makes it difficult for a reader less familiar with these structures to orient and understand the consequences of slight changes in coordination geometry. Please include the backbone ribbon in gray for both structures so that the coupled changes in side chain orientation to coordinate the ions and TM helix orientation.

-Minor quibble - Fig. 5b, e are very faint. Reducing the depth cue slightly would help make the figure more visible.”

These are very relevant comments. Figure 4 has now been substantially revised. We now show the structure in two different angles in an orientation that should allow the reader to understand the positioning within the membrane. To give a better view of the sodium sites, there are now two subfigures, one highlighting Na1 and one highlighting Na2. In addition, there is a figure highlighting the L-Phe binding site. We have also revised Supplementary Fig. 5 so it should be easier to understand, and, finally, Fig 5b is now less faint.

“Reviewer #2 (Remarks to the Author):

The manuscript under review presents a new LeuT (W8A mutant) crystal structure in a Na⁺ and L-Phe bound inward facing occluded state. The choice of mutation and bound substrate in the crystallization process is guided by smFRET and MD data indicating that the W8A substitution and L-Phe binding stabilize inward facing conformations of LeuT. In the new crystal structure important interactions between gating residues in the N-terminus and TMs 1 and 8 are altered, leading to opening of the protein toward the cytoplasm due to movement of TM5 and potentially increased hydration (as evidenced from MD simulations) of the Na2 binding site, necessary for future Na2 dissociation. Comparison to existing outward facing occluded and inward facing opened structures of LeuT, as well as 2 available inward facing occluded structures of MhsT, show that the new LeuT structure features significant changes in the position of TM5 without much change in TM1, which places it as a novel intermediate structure between the outward facing occluded and inward facing open conformations in the catalytic cycle of LeuT. The study is well executed, with appropriate methodology and the conclusions are consistent with the provided results. Good understanding of the LeuT catalytic cycle is crucial for conceptual studies on secondary transport and may have immediate impact on pharmacological approaches with implications to mental health and opioid addiction. As such, the study is both novel and significant and is fit for publication in Nature Communications after a minor revision.

The authors report MD simulations on WT and W8A LeuT models based on an existing outward facing structure (3GJD), which guide their crystallization strategy. Later, MD simulations with the crystalized LeuT (W8A, Na/L-Phe bound) monomers are also reported, where water permeation toward the Na2 site is visible. How do the structures from these two sets of calculations compare? Are there any indications of TM5 movement caused by the W8A mutation or substrate binding in the MD simulations with the 3GJD models (WT out-occ and W8A out-occ), consistent with the new crystallized structure? Considering the length of the simulations (1 microsecond per replica) and the number of replicas (10+10) even slow TM movements might be observable in some of the replicas of these ensembles. Did the hydration patterns at the end of the 1 microsecond in the 3GJD models resemble those of the new crystal structure? Did Na⁺ from Na2 dissociate in any of the MD simulations? Basically, do the simulations, starting from the outward-facing occluded models lend support to the transport hypothesis of the authors (movement of TM5, no movement of TM1, increased hydration of Na2 site) and that the new crystal structure is an intermediate in the catalytical cycle and not an artifact of the W8A mutation? A few words about this in the text of the manuscript will strengthen the drawn conclusions. Overall, very interesting and thought-provoking studies that deserves publication in Nature Communication pending minor revisions noted above.”

Again, we appreciate the favorable comments from the Reviewer. We also agree that the questions raised by the Reviewer indeed are interesting and thoughtful. The simulations we carried out are indeed relatively long and were run in several replicates, but the LeuT system is notorious for its “stiffness”. The simulations of the W8A mutant started from the occluded structure (W8A_{out-occ}), and the major dynamic changes we observe are related to the motion of the Y268 side-chain (Fig. 2b-c) as well as the increase in hydration of the intracellular channel leading to the Na2 site (Supplementary Fig. 3). On the time-scales of the simulations, the W8A_{out-occ} system samples states where the channel is populated with six water molecules on average, forming a water wire that connects the Na2 site to the intracellular vestibule (see figure below, left panel). For comparison, in the occluded state simulations this region is mostly devoid of water with only one of the replicas displaying a transiently formed water wire (Figure below, right panel). This is consistent with our previous finding that water wires in this region are transient on the microsecond scale, even when mutations are performed that stabilize the inward-facing state (Ref. 35). Overall, we estimate that we are capturing what we believe to be the initial stages of structural transformation from the starting structure of the occluded state to the intermediate state.

It is important to note that the number of water molecules in the channel of the simulated W8A_{out-occ} system is very similar to that found in the W8A_{chainA/B} systems (the simulations of the two chains in the W8A X-ray model). This is shown in Supplementary Fig. 6. Notwithstanding this consistency between the simulations starting from the two models, the time-scales of our simulations are still likely insufficient to follow subsequent steps in conformational transition from the occluded to the intermediate state, such as the process of Na2 release.

Indeed, in the large-set (50 microseconds) of MD simulations of the mammalian homologue of LeuT, the wild type human dopamine transporter (hDAT) (presented in Ref. 20), we

sampled as rare events spontaneous transitions of the transporter from the occluded state to the intermediate state in which TM5 partially unwinds. These events were related to the increase in the hydration of the Na2 site such that the Na2 ion was fully solvated by water molecules. Comparatively, the sampling of this processes in LeuT from unbiased MD simulations presents a challenge both in view of its inherently slow dynamics at ambient (room or body) temperatures, and also because it lacks the long termini segments which we have shown to play a key role in the functional dynamics of the mammalian homologue (hDAT) (see especially Ref. 44). Nevertheless, we expect the LeuT structure in the $W8A_{OUT_OCC}$ simulations to reach the intermediate state if the conformational sampling is continued adaptively as we have done for enhanced sampling of rare dynamic events in other membrane proteins (Morra et al, Structure 2018, 26, 356-367.e3; Lee et al, Nature Communications, 2018 9,3251). However, such simulations are beyond the scope of the current work as we believe the presented data lends support for the transport hypothesis by showing the size and hydration increases of the intracellular channel in the $W8A$ mutant, suggesting that the system is on path to reconfiguration of the intracellular vestibule towards the intermediate (and eventually towards inward-open) state as was seen in DAT (Ref. 20).

To further clarify these points, we have introduced the Figure shown below into the revised Supplementary Figure 3 as panels c-d, and have revised the main text of the manuscript accordingly (in Results, page 8, 1st paragraph – page 9, 1st paragraph, and in Discussion, page 16, 1st paragraph).

Figure (for panels c-d of Supplementary Fig. 3): Histogram of number of waters in the channel from the analysis of the 10 individual trajectories of the $W8A_{OUT_OCC}$ (left) and WT_{OUT_OCC} (right) systems. Water wires connecting the Na2 site to the intracellular vestibule are formed in half of the $W8A_{OUT_OCC}$ trajectories but only in one trajectory for the WT_{OUT_OCC} system (see probabilities for number of water molecules ≥ 6).

“Reviewer #3 (Remarks to the Author):

Gotfryd et al. present the previously elusive state of LeuT in sodium/substrate bound inward occluded conformation at the relatively high resolution of 2.6 Å. This is a result of a nice and elegant approach of combination of smFRET and MD simulations that led to a proposal that this state can be arrested in presence of L-Phe and W8A mutant. I have no concerns about the structure - however looking at figure S4 - there are some FoFc peaks, if the sigma of 2fo-fc map is lowered is it possible to build extra residues? If so, the further calculation of the polder maps might be helpful. It would be also nice if the authors show omit maps for sodium ions and L-Phe.”

We are pleased that the Reviewer acknowledges our work. We agree that there are some Fo-F_C peaks; however, even when lowering the sigma contour level to 0.5, we cannot reveal electron densities for additional residues, indicative of disorder of the N-terminal region in the crystals. This is now illustrated in Supplementary Fig. 4. Moreover, as requested by the Reviewer, we now show omit maps for bound sodium ions and L-Phe as part of Supplementary Fig. 4.

“Apart from the structure I am puzzled with two observations - (1) the affinity to L-Phe is lower than to L-Leu despite that L-Phe has more favourable contacts - hydrophobic and aromatic and (2) weird transport behaviour of WT and W8A mutant I think the authors should elaborate on that.”

As pointed out by the Reviewer, we find for the WT transporter an affinity of L-Leu that is markedly higher than of L-Phe. Importantly, this is consistent with previous findings for aromatic amino acid binding to LeuT (Ref. 39). It is important to note that the affinity is not only determined by the number of favorable contacts but also by the propensity of the binding pocket to assume the most favorable conformation. In Ref. 39, LeuT was crystallized in an outward-open configuration in the presence of the analog L-4-F-Phe. Interestingly and consistent with lower affinity of this compound as compared to L-Leu, the structure revealed a strained configuration in agreement with a less favorable conformation of the binding pocket in the outward-facing occluded state for aromatic amino acids.

In the W8A mutant, we find a dramatically lowered affinity of L-Leu (~1600 nM versus ~20 nM in WT) while the affinity of L-Phe is essentially the same (800 nM versus 900 nM) and thereby slightly higher than that found for L-Leu. We argue that this agrees with W8A assuming a more inward facing configuration with a less favorable conformation of the binding pocket for L-Leu binding. In contrast, L-Phe binding affinity is preserved suggesting that this more inward facing configuration is relatively more favorable for accommodating L-Phe – and possibly, therefore, we can stabilize and crystallize W8A in the present state.

As for transport behavior, we observe that [³H]Ala, but not [³H]Phe, is a substrate for WT. This is consistent with the smFRET data, suggesting that L-Phe stabilizes an intermediate-FRET state and that the L-Phe-bound WT displays no apparent transitions between high- and low-FRET states as it is seen for [³H]Ala. When mutating W8, transport of [³H]Ala is lost,

suggesting that release of the N-terminus and destabilization of the inner gate impairs the ability of L-Ala to promote transport. We have no immediate explanation for the very low, yet detectable transport, that we observe for [³H]Phe in W8A but it is tempting to suggest that it reflects a more favorable interaction of L-Phe with the inward facing W8A conformation.

The topic is now discussed on page 15 - page 16, 1st paragraph.

“The minor things: reaction cycle intermediates - I believe transport cycle intermediates is more appropriate.”

This has now been changed so we say “transport cycle intermediates” and not “reaction cycle intermediates.”

“Please calculate the volume of enlarged cavity.”

This is yet another good suggestion from the Reviewer. Thus, to compare quantitatively the extent of opening of the intracellular vestibule in the W8A structure to that in the occluded conformation of LeuT, we calculated the number of water molecules in the intracellular vestibule (defined identically for the two structures) during the course of MD simulations of the respective systems (see Methods). As shown in a new Supplementary Fig. 7, the volume of the intracellular vestibule in the W8A structure fits on average ~30 waters, whereas in the occluded LeuT, on average, only ~10 water molecules populate this region. The difference in the local structure of the W8A and the outward-facing occluded state of LeuT is illustrated in Supplementary Fig. S7 by a comparison of the positions of C α atoms of residues located at the intracellular entrance to the vestibule. The structural changes related to the movement of the TM5 and the release of the N-terminus are seen to result in widening of the intracellular vestibule.

The result is described in Results on page 13, 1st paragraph

“Also maybe worth discussing whether such unwinding of helices (discussion) is feasible in membrane?”

We thank the Reviewer for pointing out the need for addressing this issue. Indeed, transmembrane helix (TM) unwinding is not uncommon, both structurally and in the context of functional dynamics. A pertinent example is the state-dependent partial unfolding of TM5 observed in the MD simulations of the hDAT in explicit membrane environment (Ref. 20). In addition, the unwinding seen in LeuT by application of HDX-MS (hydrogen deuterium exchange mass spectrometry) in detergent solution was also observed upon reconstituting LeuT into nanodiscs and thus a lipid membrane environment (Ref. 32).

The topic is now briefly discussed in Discussion on page 19, 2nd paragraph

Reviewers' Comments:

Reviewer #1:

Remarks to the Author:

The additional experiments, explanations and figure revisions made by the authors have addressed all of my concerns. This is now a very strong manuscript that is suitable for publication in Nature Communications.

Reviewer #2:

Remarks to the Author:

The paper has been revised and considerably. I appreciate the authors attentiveness and ability to constructively address my criticism. I am pleased to recommend its publication in the current form.

Reviewer #3:

Remarks to the Author:

The authors have addressed all the issues, the manuscript has become even better so I fully support its publication in Nature Communications